# Norwegian Atlantic Slope Current along the Lofoten Escarpment

Ilker Fer[1], Anthony Bosse[1,2], and Johannes Dugstad[1]

[1]Geophysical institute, University of Bergen, and Bjerknes Centre for Climate Research, Bergen, Norway
[2]Now at Aix-Marseille Univ., Université de Toulon, CNRS, IRD, MIO UM 110, Marseille, France

**Correspondence:** Ilker Fer (ilker.fer@uib.no)

**Abstract.** Observations from moored instruments are analyzed to describe the Norwegian Atlantic Slope Current at the Lofoten Escarpment (13°E, 69°N). The data set covers a 14-month period from June 2016 to September 2017, and resolves the core of the current from 200 to 650 m depth, between the 650 m and 1500 m isobaths. The along-slope current, vertically averaged between 200 and 600 m depth has an annual cycle amplitude of $0.1~\mathrm{m\,s^{-1}}$ with strongest currents in winter, and a temporal average of $0.15~\mathrm{m\,s^{-1}}$. Higher frequency variability is characterized by fluctuations that reach $0.8~\mathrm{m\,s^{-1}}$, lasting for 1 to 2 weeks, and extend as deep as 600 m. In contrast to observations in Svinøy (2°E, 63°N), the slope current is not barotropic and varies strongly with depth (a shear of 0.05 to $0.1~\mathrm{m\,s^{-1}}$ per 100 m in all seasons). Within the limitations of the data, the average volume transport of Atlantic Water is estimated at $2.0\pm0.8$ Sv ($1~\mathrm{Sv} = 10^6~\mathrm{m^3\,s^{-1}}$), with summer and winter averages of 1.6 and 2.9 Sv, respectively. The largest transport is associated with the high temperature classes ($> 7°\mathrm{C}$) in all seasons, with the largest values of both transport and temperature in winter. Calculations of the barotropic and baroclinic conversion rates using the moorings are supplemented by results from a high resolution numerical model. While the conversion from mean to eddy kinetic energy (e.g. barotropic instability) is likely negligible over the Lofoten Escarpment, the baroclinic conversion from mean potential energy into eddy kinetic energy (e.g. baroclinic instability), can be substantial with volume-averaged values of $(1-2) \times 10^{-4}~\mathrm{W\,m^{-3}}$.

## 1 Introduction

The relatively mild climate of Norway is largely attributed to the northern extension of the North Atlantic Drift, the Norwegian Atlantic Current that transports warm and saline water masses toward the Arctic Ocean (Seager et al., 2002; Rhines et al., 2008). These nutrient-rich warm waters contribute to support the entire food chain and sustain the productive waters around Norway (see e.g., Sundby (2000) for a discussion on recruitment of Atlantic cod stocks). The circulation pattern is organized in two main branches originating from the Iceland-Faroe and Faroe-Shetland gaps (Poulain et al., 1996; Orvik and Niiler, 2002) (Fig. 1a): the Norwegian Atlantic Slope Current (the slope current hereinafter) and the Norwegian Atlantic Front Current (the front current hereinafter). The diverging isobaths of the Lofoten Basin in the Norwegian Sea guide the two branches. The slope current follows the shelf break along the Norwegian continental slope northward and continues into the Barents Sea and Fram Strait. The front current follows the 2000-m isobath, veers west at the flanks of Vøring Plateau and continues poleward along the Mohn Ridge (Orvik and Niiler, 2002; Bosse and Fer, 2019).

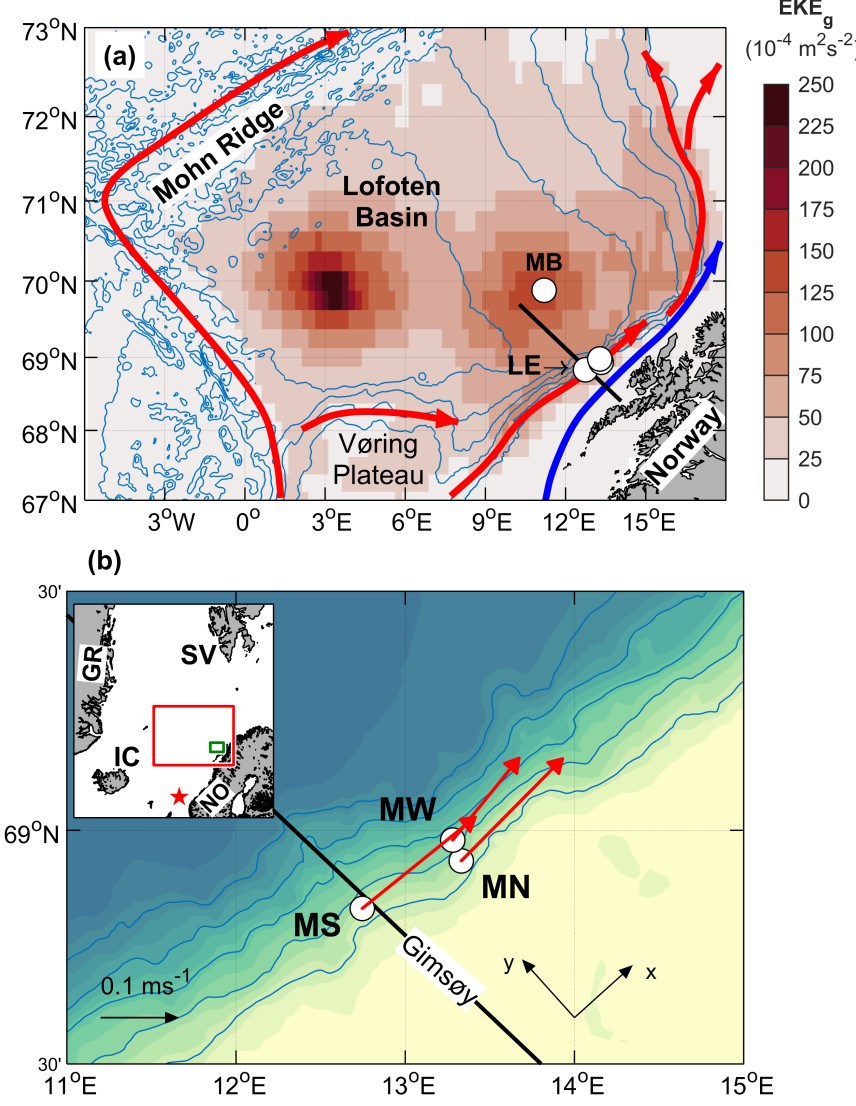

**Figure 1.** (a) Bathymetry of the Lofoten Basin in the Norwegian Sea (ETOPO1, contours at 500 m intervals) and the geostrophic EKE (EKE$_g$) averaged over the period 1993 to 2018, calculated using sea-level anomalies from satellite altimeter observations. General circulation of the warm Atlantic Water is indicated by red arrows showing the slope current and the front current. The Norwegian coastal current is indicated by the blue arrow. Black transect is the portion of Gimsøy section shown in Fig. 2. Mooring positions are shown by circles, also showing the basin mooring (MB) at the secondary EKE$_g$ maximum. Lofoten Escarpment (LE) is the steep slope region near the slope moorings. (b) A zoom-in to the moorings analyzed in this study, showing MS, MN and MW together with 200–600 m depth-averaged current vectors (scale on lower left), Gimsøy section (black), and the orientation of the coordinate system (along-slope, $x$, and across-slope, $y$). Blue isobaths are drawn every 500 m. The inset is a location map with domains of (a) and (b) marked in red and green, respectively. The monitoring location for the Svinøy section is shown by the red star. NO: Norway, SV: Svalbard, IC: Iceland, GR: Greenland.

The front current, which is not addressed in this study, has not been measured in detail using current meter arrays, but geostrophic transport estimates are available from hydrography. At the Svinøy section (63°N, about 300 km downstream of the Faroe-Shetland Channel), a baroclinic geostrophic transport estimate of the front current was 3.4 Sv (1 Sv = $10^6$ m$^3$ s$^{-1}$) (Orvik et al., 2001); however, the total geostrophic transport from repeated Seaglider transects reached 6.8 Sv (Høydalsvik et al., 2013), implying a large barotropic contribution. Farther north, detailed glider observations of the front current over the Mohn Ridge confirm large transport rates giving 4.5 Sv annual average with approximately 2 Sv barotropic contribution (Bosse and Fer, 2019).

University of Bergen, Norway, has monitored the slope current transport at the Svinøy section since 1995, at the location indicated by a star in the inset of Fig. 1b (Orvik et al., 2001). The slope current there is about 40 km wide, between the 200 and 900 m isobaths, with an annual mean speed of 0.3 m s$^{-1}$. The average annual transport of this barotropic branch is 4.4 Sv (Orvik et al., 2001; Orvik and Skagseth, 2003). The slope current accelerates along steep topography off the Lofoten Escarpment near the Lofoten Islands (Poulain et al., 1996). The Norwegian coastal current (blue arrow Fig 1), carries relatively fresh water over the shelf and as the shelf gets narrow near the Lofoten Escarpment, there might be interactions with the slope current. Here, there are no published moored current meter records, but surface drifters indicate velocities reaching 1 m s$^{-1}$ (Andersson et al., 2011). The transport and variability of the slope current in this region is not known. It is hypothesized that the current becomes increasingly unstable near this topographic steepening. Using time-averaged fields of an eddy-resolving numerical ocean simulation, Isachsen (2015) showed that the steep Lofoten Escarpment exhibits enhanced unstable baroclinic growth rates and large velocity variability, suggesting high lateral diffusion rates. The structure and transport of the slope current at the Lofoten Escarpment is the focus of this study.

The Lofoten Basin is affected by the Atlantic Water (AW) transport, and becomes a major heat reservoir that is exposed to large surface heat losses (Rossby et al., 2009b; Dugstad et al., 2019a) and substantial water mass transformations (Rossby et al., 2009a; Bosse et al., 2018). The AW enters the basin both as a broad slab in the upper layers between the two branches (Rossby et al., 2009b; Dugstad et al., 2019a) and by eddies detached from the unstable slope current (Köhl, 2007; Isachsen, 2015; Volkov et al., 2015; Richards and Straneo, 2015). The eddy-induced lateral heat fluxes distribute the heat in the basin (Spall, 2010; Isachsen et al., 2012; Dugstad et al., 2019a). The region is energized, manifested in the map of average geostrophic eddy kinetic energy (Fig. 1a, see Sect. 3) showing two maxima: one in the center, associated with a permanent, energetic eddy (Ivanov and Korablev, 1995; Søiland and Rossby, 2013; Fer et al., 2018; Bosse et al., 2019), and a secondary maximum closer to the slope, likely associated with the variability induced by the slope current. The energetics and the variability of the slope current remain to be constrained by observations.

The study was conducted as a part of the "Water mass transformation processes and vortex dynamics in the Lofoten Basin of the Norwegian Sea" (PROVOLO) project. The overall objective of PROVOLO was to describe and quantify the processes and pathways of energy transfer and mixing in the Lofoten Basin and their role in water mass transformation. Observations from multiple cruises, gliders and subsurface floats were analyzed and reported elsewhere with focus on AW transformation (Bosse et al., 2018), the permanent Lofoten Basin Eddy (Fer et al., 2018; Bosse et al., 2019) and the frontal structure across the Mohn Ridge (Bosse and Fer, 2019). The mooring component concentrated on the slope current. Here we report the first

**Table 1.** Mooring deployment and recovery details. Total depth is estimated from the deepest pressure sensor, mooring line construction and the ship's echo sounder.

| Mooring | Latitude | Longitude | Depth (m) | Deployed | Recovered |
|---------|----------|-----------|-----------|----------|-----------|
| MS | 68°N 50.038′ | 012°E 44.777′ | 672 | 31 May 2016 | 08 Sep 2017[a] |
| MN | 68°N 56.109′ | 013°E 19.866′ | 655 | 01 Jun 2016 | 08 Sep 2017[b] |
| MW | 68°N 58.759′ | 013°E 16.845′ | 1500 | 01 Jun 2016 | 08 Sep 2017 |
| MB | 69°N 52.89′ | 011°E 11.89′ | 2925 | 02 Jun 2016 | 09 Sep 2017 |

[a] Water column line is lost

[b] Water column line is recovered on 24 Aug 2016

observations of the volume transport rates, energetics and their variability from weekly to seasonal time scales based on the mooring records.

## 2  Data

### 2.1  Moorings

A set of 4 moorings was deployed across the continental slope of the eastern Lofoten Basin (Fig.1). A deployment and recovery summary is listed in Table 1, and full details are provided with the documentation following the data set (Fer, 2020). Mooring name convention is Mooring North (MN), South (MS), West (MW) and Basin (MB). MB was located at the secondary geostrophic EKE maximum (Fig. 1a) to address the mesoscale variability in the basin. Data from this mooring will be analysed for a separate study and are not reported here. The observations cover a 14-month period from June 2016 to September 2017.

The arrangement of the three moorings on the slope (Fig. 1b) was designed to cover the core of the slope current (two moorings at the 650 m and 1500-m isobaths, MN and MW), and to investigate the co-variability along the slope (two moorings at the 650-m isobath). The along-slope distance between MS and MN is 26 km, and the cross-slope distance between MN and MW is about 5 km. Moorings MS and MN at the 650 m isobath each consisted of one bottom unit and a water column line with distributed conductivity-temperature-depth (CTD) sensors. The bottom units were approximately 25 m tall, equipped with a RDI 75kHz Longranger acoustic Doppler current profiler (ADCP) in a spherical flotation and a Sea-Bird Scientific SBE37 Microcat with CTD sensors. This approach mitigated the high risk due to fisheries activities. The ADCP bottom unit and mooring line pairs were deployed close to each other at approximately the same isobath (within 5 m), and within 250 m horizontally, and will be treated as a single mooring. Unfortunately both water column mooring lines at MN and MS were damaged by fishing boats. The MS line was lost with no data return. The MN line was cut after 3 months. The drifting MN line together with the sensors were recovered, giving 3 (summer) months of temperature and salinity data in the water column. The current profile and the near-bottom CTD data from the bottom units were successfully recovered and cover the whole study period.

The moorings were densely instrumented and sampled at a hourly rate or faster, covering a large fraction of the water column. The instrument target depths can be seen on the vertical axis of Fig. 4, introduced later. Currents were measured using ADCPs, mainly RDI 75kHz Longrangers for the moorings reported here, and point current meters (Aanderaa SeaGuard, Xylem Inc.). The ADCPs on MN, MS and MW were placed to cover the dynamic core of the slope current (at approximately 10 m height above seabed at the 650 m isobath, and at 740 m depth at the 1500 m isobath, each pointing upward with about 550 m range). Temperature, salinity and pressure were sampled using SBE temperature loggers (SBE56 and SBE39) and CTD recorders (Microcat, SBE37). Detailed instrument distribution on moorings can be found in the data set documentation (Fer, 2020). Current measurements were corrected for magnetic declination. After all moorings were recovered, a calibration CTD cast was made with all mooring SBE sensors attached to the ship's CTD frame. The temperature and salinity measurements were corrected to be internally consistent, and also against the calibration cast and the profiles taken when the moorings were in water. Applied offset corrections for each instrument are listed in the data set report, and vary in the range of $(1-40)\times10^{-3}\,°C$ for temperature and $(2-50)\times10^{-3}$ for practical salinity.

Substantial vertical displacements ("knock-down") of the mooring line occurred in response to strong current events at MW and MB (not reported here). At MW, the vertical displacements recorded by the uppermost pressure sensor at 75 m target depth was 7 m (50 percentile), 15 m (80 percentile, corresponding to a total duration of about 3 months) and 68 m (97 percentile, corresponding to events with a total duration of 2 weeks). The vertical displacements were reduced by approximately a factor of two at the level of the ADCP flotation at 740 m depth. The velocity measurements from the ADCPs installed in the bottom units at MN and MS were relatively unaffected by the mooring motion (typical vertical displacements associated with knock-down were less than 1 m with a 97 percentile value of 2 m). Overall, the moorings were equipped with several pressure sensors which we used to approximate the depth of temperature, salinity and current measurements in the water column.

A data set was prepared after correcting for mooring knock-downs. Data from all instruments were first averaged into one hour intervals (if the sampling rate was faster) and then interpolated to a common 1-hour time array. Time series of instrument depth were constructed at each time and for each instrument using vertical interpolation of the known target depth (of instruments with pressure sensor) and the measured pressure to the target depths of all instruments. Hourly profiles of temperature, salinity and horizontal current were then vertically interpolated to a uniform 10-m vertical resolution. Data gaps at a given vertical level were typically caused by mooring knock-down or lack of acoustic scatterers for Doppler velocity measurements. At MW, velocity measurements were relatively limited in the vertical. The data gap in the time series was 18% at 250 m depth, reaching 60% at 200 m. The vertical extent of temperature measurements at MW was better: temporal gap at 90 m was only 20%, increasing to 70% at 80 m. The missing velocity data at MN was 35% at 150 m, increasing to 50% at 80 m. A depth level with a data coverage less than 30% of the total measurement duration was excluded from the data set.

The initial accuracy of the SBE sensors are $\pm2\times10^{-3}\,°C$ for temperature, $\pm3\times10^{-4}\,S\,m^{-1}$ for conductivity, and $\pm1$ dbar for pressure (drift over 1 year is comparable to initial accuracy for temperature and pressure, and 10 times the initial accuracy for conductivity). For the deployment setup used, the ADCPs have a single ping (profile) statistical error of $2.5\,cm\,s^{-1}$, which reduces to $0.4\,cm\,s^{-1}$ for the ensemble average profile with 35 pings. The compass direction is accurate to $\pm2°$. Conservative error estimates are $\pm1\,cm\,s^{-1}$ for velocity, $\pm10^{-2}\,°C$ for temperature and $\pm10^{-2}$ for practical salinity.

For the analysis in this study, we rotated the coordinate system by $42°$ from East, with $x$-axis pointing along-slope and $y$-axis cross-slope toward deeper water (see Fig. 1b). Mean orientation of the slope was calculated using isobaths from ETOPO1 near the slope moorings. Current components are along-slope, $u$, and across-slope, $v$. Hourly-averaged data set was filtered using a 14 day low-pass filter for background fields, and 35 h to 14 day band-pass filter for eddy covariance and conversion rate calculations. In both cases a 3rd order phase-preserving Butterworth filter was used.

## 2.2 Other data

Atmospheric forcing was obtained from European Centre for Medium-range Weather Forecast (ECMWF) (2011) ERA-Interim (Dee et al., 2011) reanalysis over the historical time period from 1979 to 2018, and from higher resolution ERA-5 reanalysis (Copernicus Climate Change Service (C3S), 2017) over the mooring observation period. Surface net fluxes $Q_{net}$ (downward positive) were computed as the sum of net shortwave and longwave contributions and latent and sensible heat fluxes. Time series of fluxes were extracted at the nearest grid point from mooring sites. We calculated the geostrophic EKE ($EKE_g$) using the surface geostrophic velocity anomalies obtained from sea-level anomaly from the multimission altimeter satellite gridded sea surface height observations distributed by E.U. Copernicus Marine Service Information.

Hydrography data from the standard Gimsøy section available from 4 occupations during the mooring period (on 30 Jul 2016, 19 Nov 2016, 8 Mar 2017 and 7 Jun 2017) were obtained from the Norwegian Marine Data Center.

Climatological transects at the Svinøy and Gimsøy sections were constructed from a hydrographical Atlas of the Nordic Seas (Bosse and Fer, 2018). This is a merged data set including observations from shipboard CTDs, Argo profiling floats and underwater gliders between 2000 and 2017. To construct the sections discussed in Sect. 7, we used all profiles located within 25 km distance to the Svinøy and Gimsøy transects, projected horizontally onto the transect and binned in 5 km cross-section intervals. Seasonal averages for temperature and salinity were smoothed using a Gaussian moving window of 10 km variance. Finally, we calculated the annual mean by averaging over four seasonal sections.

In order to assess how representative our discussion of energetics obtained from mooring data is of the volume-averaged energetics in the region, we performed calculations using outputs from a high-resolution Regional Ocean Modelling System (ROMS) configuration in the Nordic Seas. ROMS is a hydrostatic model with terrain-following coordinates that solves the primitive equations on a staggered C-grid (Shchepetkin and McWilliams, 2009; Haidvogel et al., 2008). The model outputs used here have a horizontal resolution of 800 m, 60 vertical layers with increased resolution towards the surface (1-3 m at the surface, about 60 m at the bottom) and are stored as 6 hourly outputs. The model fields are described in detail in Dugstad et al. (2019b).

## 3  Context and environmental forcing

The standard Gimsøy section was visited 4 times throughout the mooring period. An average section using the subset of stations taken in all 4 occupations is representative of the hydrography during the measurements (Fig. 2; also compare to the section from climatology presented in Sect. 7). The AW, identified by temperatures above $5°$ C and absolute salinities $S_A > 35.17$,

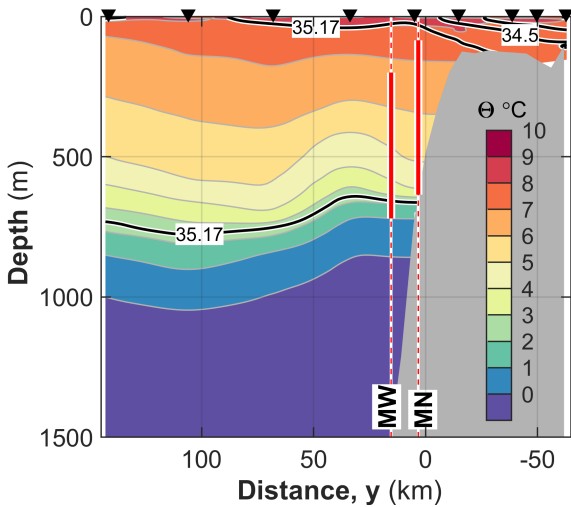

**Figure 2.** Conservative Temperature ($\Theta$, in color, contours at 1°C interval) and Absolute Salinity (black lines, 34, 34.5 and 35.17 g kg$^{-1}$ contours) distribution at the Gimsøy section averaged over four occupations throughout the mooring period. Only stations (arrowheads) with four profiles are used. $S_A$ = 35.17 approximately corresponds to a practical salinity of 35 (typical lower limit for AW) at this latitude, 300 dbar pressure and 5°C temperature. Bathymetry is from ETOPO1. Distance along $y$ is referenced to the 500 m isobath. The moorings MN and MW are shown at the distance on the section corresponding to their deployment isobaths. The vertical extent of the ADCP current profiling is marked with thick red.

covered the 50–700 m layer from the shelf edge toward the basin, overlying the fresher and colder deeper water. The interface between these water masses meets the bottom slope at about 700 m. Relatively fresh layer on the shelf is associated with the Norwegian coastal current. The moorings MN and MW, marked on Fig. 2, show that the range of current measurements sufficiently covered the AW layer and the dynamical core at the slope identified by sloping isotherms.

A summary of the environmental forcing during the measurement period shows that the net surface flux was typical of the long-term average, with an event of strong heat loss exceeding 1 standard deviation ($\sigma$) envelope from mid February to early March 2017 (Fig. 3a). Wind speed showed a seasonal variability increasing from 5 m s$^{-1}$ in summer to 12 m s$^{-1}$ in winter (Fig. 3b). We averaged the EKE$_g$ from satellite altimetry in a 30-km radius at the basin mooring location (a EKE$_g$ maximum region, see Fig. 1a) and at MW, and compare the evolution throughout the mooring deployment in Fig. 3c. The EKE$_g$ records confirm that MB is 2 to 5 times more energetic in general, except in summer when both locations were relatively quiescent.

## 4 Average properties and seasonal profiles

Profiles temporally averaged in the winter (DJF), spring (MAM), summer (JJA) and fall (SON) months at MW and MN show strong vertical shear in $u$ in the upper 600 m at both moorings (Fig. 4). In contrast to the barotropic slope current at Svinøy, the current at the Lofoten Escarpment clearly has a strong baroclinic component. Background shear between 200 and 600 m

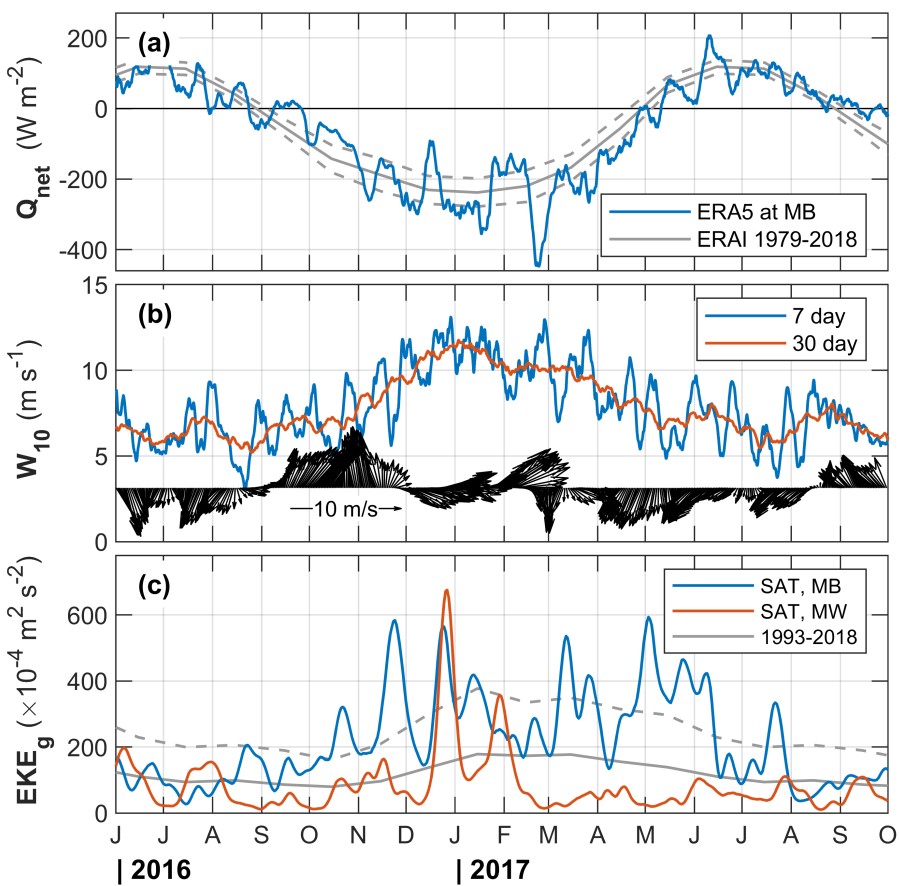

**Figure 3.** Environmental forcing conditions throughout the mooring deployment period. a) Net surface heat flux, $Q_{net}$ from ERA5 at the grid point closest to MB, together with monthly average and one standard deviation ($\sigma$) envelope for the period between 1979 and 2018. b) Weekly and monthly averages of 10-m wind speed and wind vectors from ERA5. c) $EKE_g$ from satellite altimetry calculated at the grid point close to MB (blue) and MW (red), together with monthly average and one $\sigma$ between 1993 and 2018 at MB.

depth was 0.05 to 0.10 $m\,s^{-1}$ per 100 m in all seasons, with a maximum in the fall. The fall was characterized by maximum

baroclinicity, whereas winter was characterized by maximum barotropic currents, consistent with increased winds. Increased baroclinicity in the fall could partly be due to seasonal freshening of the coastal current reinforcing the density gradients, and partly due to increased Ekman transport toward the shore observed from September to March (see northward winds implying eastward Ekman transport in Fig. 3b). It is also likely that the slope current could interact with the fresh coastal current due to the narrow shelf off the Lofoten Escarpment.

Over the full record, 200–600 m depth average $u$ was 0.15 $m\,s^{-1}$. The strongest currents were observed in winter with an average of 0.20 $m\,s^{-1}$ at MW and 0.25 $m\,s^{-1}$ at MN (approximately twice the summer average) when the temperature was also the largest. Average winter temperature at MW was 7.3°C, compared to 5.8°C (at MW) or 6°C (at MN) in summer. The

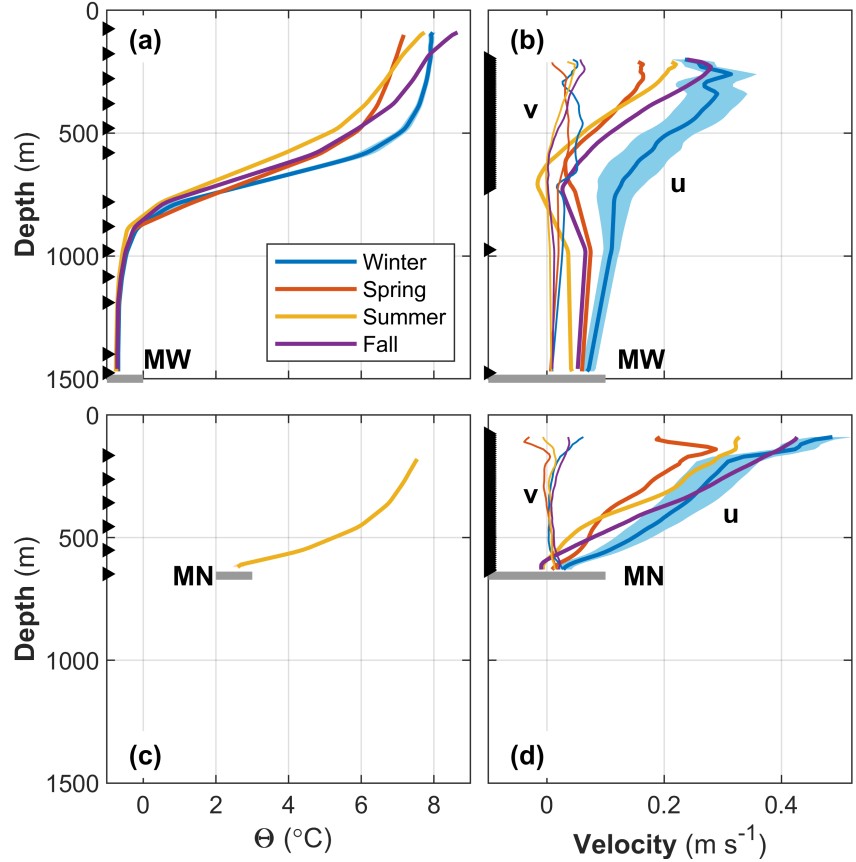

**Figure 4.** Time averaged profiles of (a,c) $\Theta$ and (b,d) velocity components $u$ (thick) and $v$ (thin lines) for moorings MW (upper row) and MN (bottom row). Time averaging is made over seasons, winter, spring, summer and fall, as indicated in the legend. Gray horizontal line marks the sea bed. Arrowheads on the vertical axis mark the target depth of measurements. The error bars are the standard error using a decorrelation time scale of 7 days, for clarity shown only for winter (summer for MN temperature) and are comparable in other seasons. The error shading for temperature is not distinguishable from the average profile.

average temperature in the 200–600 m range was warmer by more than 1°C in winter. This could partly be explained by the vertical redistribution by winter vertical mixing of heat contained in the seasonal thermocline, and partly by changes in AW properties flowing into the Nordic Seas. Cross-slope component was weak (typically $\pm 0.02$ m s$^{-1}$) and increased in spring and winter, with largest 200-600 m depth-averaged values in winter (0.05 m s$^{-1}$ at MW) and an increased variability with depth (Fig. 4b, d). In the upper part of the water column at MW, averaged cross-slope velocities in fall exceeded the winter values. This is consistent with the increased EKE$_g$ at MW location, calculated from satellite measurements in November 2016 (Fig. 3c). In deep layers (>900 m) at MW, barotropic currents were between 0.05 and 0.10 m s$^{-1}$.

The summer profiles in Fig. 4 are averages over summers of 2016 and 2017. When averaged separately (not shown), temperature profiles at MW are very similar, equal to within $0.5°C$ in the upper 600 m and identical in deeper layers. At MW, $u$ was $0.01\,\mathrm{m\,s^{-1}}$ larger below 300 m (a small barotropic increase) in summer 2017, and shear was stronger in the upper 300 m, increasing by $0.06\,\mathrm{m\,s^{-1}}$ to 200 m depth. At MN, summer-average profiles of $u$ in the bottom 250 m were identical in 2016 and 2017, but shear was stronger higher in the water column (above 400 m) in summer 2017, with $u$ increasing by an additional $0.10\,\mathrm{m\,s^{-1}}$ to 200 m depth. This implies a substantial inter-annual variability in the upper $300 – 400$ m which cannot be resolved by our limited times series.

## 5  Temporal variability

The currents measured at moorings MW and MN were highly variable (Fig. 5). The 14-day low-passed currents were strongest in the fall and winter (Fig. 5a-d). The annual cycle of the 200–300 m vertically averaged $u$ at MN had an amplitude of 0.10 $\mathrm{m\,s^{-1}}$ and explains 20% of the variance, obtained using a sinusoidal fit to daily data (not shown). These figures are similar for MW for 300-400 m averaged currents (depth ranges are chosen to ensure continuous time series, unaffected by mooring knockdowns). The cross-slope components show a less pronounced seasonality with 0.01-0.02 $\mathrm{m\,s^{-1}}$ (5-15% variance explained) at both moorings. Temperature record at MW also shows strong seasonality. The amplitude of annual sinusoidal fit to the temperature time series, increases from $0.6°C$ at 200-300 m to $1°C$ at 500-600 m, accounting for 60-70% of the variance, and rapidly decays deeper.

The largest along-slope currents reach 0.8 $\mathrm{m\,s^{-1}}$ at both moorings, last for 1 to 2 weeks and extend as deep as 600 m. In periods with strong $u$, the cross-slope velocity is also energized. These energetic periods also correspond with the peaks in $\mathrm{EKE}_g$ obtained from satellite altimetry at the MW location (Fig. 3c). Isotherms (available only at MW for the entire mooring record) show vertical displacements of order 100 m, consistent with mesoscale meandering of the slope current.

For comparison, in the Svinøy section Skagseth and Orvik (2002) showed that the fluctuations of the slope current are a combination of longer periodic forced oscillations which are a direct response to the wind (periods in the 3–5 day and 16–32 day bands), and free waves corresponding to the first and second topographic wave modes (dominant periods of 40–70 and 80–110 h).

We analyzed fluctuations in the low-passed fields, relative to the annual cycle, to assess dominant time scales and amplitudes of variability. The time series of fluctuations of $u$ at MN averaged between 200–300 m shows 13 events with peak-to-peak amplitude of 0.2-0.3 $\mathrm{m\,s^{-1}}$, with a mean duration of $8\pm2$ days, at an average interval (time separation between events) of $35\pm10$ days. Similar number of events with comparable time scale are detected for temperature oscillations exceeding $0.5°C$.

At shorter time scales, the 35 h to 14 day band-passed variability is shown in Fig. 5e-f, for $u$. The structure of band-passed $v$ is very similar (not shown) with approximately half the amplitude of $u$. The band-passed fields show highly energetic current variability reaching $\pm0.4\,\mathrm{m\,s^{-1}}$ (variability for $v$ is $\pm0.2\,\mathrm{m\,s^{-1}}$). A similar event analysis of the fluctuations in the filtered band (averaged between 200–300 m at MN and 300–400 m at MW) results in very similar properties for MN and MW. Typically, 40-50 events are detected in $u$ with peak-to-peak amplitude of 0.15-0.20 $\mathrm{m\,s^{-1}}$, with a mean duration of $2\pm1$ days, at an average

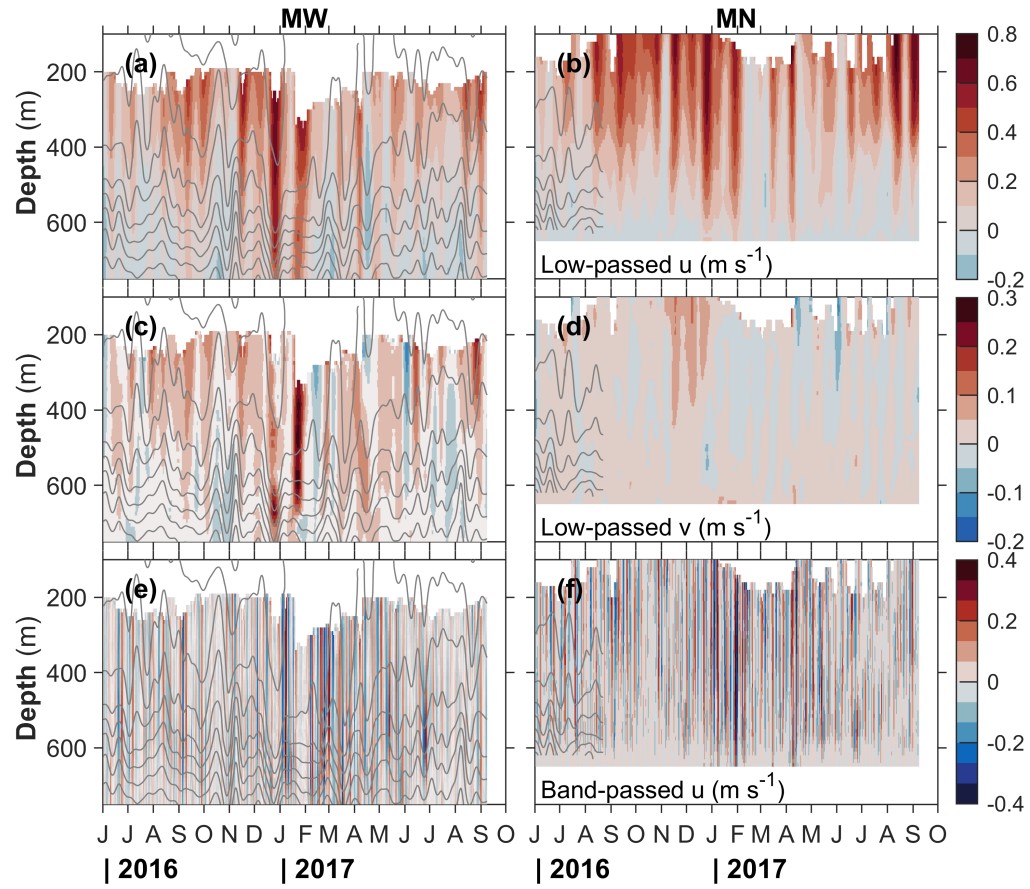

**Figure 5.** Depth-time variability of observed currents in the 100-700 m depth range at (left column) MW and right column (MN). The rows are (a,b) low-passed along slope current, (c,d) low-passed cross-slope current, and (e,f) band-passed along-slope current. The structure of the band-passed cross-slope (not shown) is similar with approximately half the amplitude. The variability in the deeper parts of MW is small and not shown for ease of comparison with MN. Isotherms at $1°C$ intervals are shown in gray in all panels. Note the lack of water column temperature data after the first three months at MN.

interval of $10\pm7$ days. The cross-slope component shows about 40 events with peak-to-peak amplitude $0.10\pm0.03\,\mathrm{m\,s^{-1}}$ at similar duration and time intervals. The energetics and conversion rates are further discussed in Sect. 8.

We estimated a de-correlation time scale as the e-folding time scale from an exponential fit to the auto-correlation function from hourly velocity time series. At both moorings at the 650-m isobath (MN and MS), 200–600 m depth-averaged along-slope currents are correlated at time scales up to 6 days. The de-correlation time scale at MW is comparable (7.3 days). For reference, advection time between the along-slope separation of MS and MN is 2 days using the mean speed of $0.15\,\mathrm{m\,s^{-1}}$. Over the 26 km separation, $u$ at MN and MS are highly coherent with a maximum correlation coefficient of $r = 0.6$ at 41 h lag (consistent

with the 2-day advection time scale). The cross-slope components are not significantly correlated. The lateral separation of 5

km between MN and MW is comparable to the Rossby deformation radius, and here $u$ is highly coherent ($r = 0.9$ at 8 h lag), and the cross-slope components are fairly correlated with $r = 0.24$ at 2-day time scale with MN leading.

## 6   Transport

Transport calculations were made using daily-averages of the 14 day low-passed current and temperature fields from moorings MW and MN, using the along-slope component of the current. First, transport densities (i.e. transport in a water column with 1 m width) were calculated by integrating vertically between 50 m and 650 m depth, roughly corresponding to the AW layer. We extended the shallowest available measurement upward to 50 m, where near-surface data are missing. The gaps in the velocity and temperature profiles vary between the moorings and are summarized in Sect. 2. A total transport was then estimated by assigning a constant width for each mooring (12.5 km for the outer mooring MW, and 7.6 km for the inner mooring MN, justified below). Positive (northeastward, $Q_p$) and negative (southwestward, $Q_n$) transport and the net transport ($Q_p + Q_n$) in $1°C$ temperature classes were computed. We estimated the transport of Atlantic Water ($Q$) as the net transport of water warmer than $5°C$. Average transports over the entire record, over summer months and winter months are listed in Table 2. Results are summarized in Fig. 6.

The moorings MN and MW are separated by 5.3 km (horizontal distance between the locations), and when projected onto the cross-slope section the distance is about 5 km (the relative angle between the mooring line orientation and the cross-isobath direction is $20°$). We assume velocity measured at each mooring is representative for the half-width (2.5 km) to the next mooring. We further extend the width of MW 10 km off-slope (distance to the 2500 m isobath) and MN 10 km on-shore (distance to the 250 m isobath). These choices are motivated by the coverage of the dynamic AW core at Gimsøy section (see Fig. 2). The outer edge corresponds to the location where the $5°C$ isotherm is shallowest, and covers the relatively steep lateral isopycnal gradient toward the slope. The width of water column for the outer mooring MW used in transport calculation is then $10 + 2.5 = 12.5$ km. The width of inner mooring is $2.5 + 5.1 = 7.6$ km, where 5.1 km is an effective width accounting for the shallowing bottom in the 10 km onshore of MN. The resulting cross-section area (600 m $\times$ 7.6 km) is equivalent to the area between 50 m and 650 m depth obtained by integrating the actual topography to 10 km onshore of the 650 m isobath.

The choice of total width for the transport calculation is consistent with the lateral structure of the depth-integrated geostrophic current inferred from the Gimsøy hydrographic section. From the 4 occupations of the Gimsøy section, we calculated the geostrophic transport relative to surface pressure. Depth-integrated geostrophic current peaks at an isobath between 500 and 750 m, suggesting that MN and MS are positioned near the maximum velocities of the slope current. The lateral structure of the depth-integrated relative geostrophic current was fairly symmetric and reduced to 20% of its maximum over a total width of 25–30 km. This lateral structure is also consistent with the vertically integrated geostrophic shear from annual mean climatology discussed later (red contours in Fig. 9a). As a result we find that the choice of cross-slope width extending between 2500 and 250 m isobaths for transport calculations is justified. Two moorings closely spaced over the slope cannot resolve the full dynamics of the slope boundary current. However, the comparison with the Gimsøy section suggests that the dynamic core of the slope current can be captured by the mooring records. The individual occupations of the section show that the bulk of

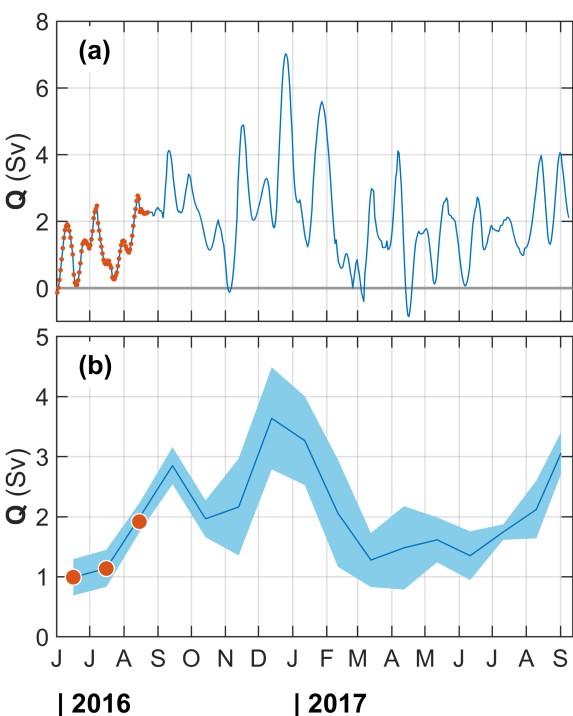

**Figure 6.** Total AW transport, in the 50–650 m range covered with moorings MN and MW, assigning a width of 12.5 km to MW and 7.6 km to MN (see text for details and sensitivity to choices). AW is defined with $\Theta > 5°C$, as measured at MW which has temperature sensors throughout the deployment. MN has temperature record in the first 3 months and the transport calculation using those records (red) agree with MW. a) Daily transports from 14 day low-passed records, b) monthly averages of the transports shown in (a). The envelope is $\pm$ standard error using a decorrelation time scale of 7 days).

the AW is in the upper 650 m, which is resolved by our moorings. The relative geostrophic transport for AW calculated in the
Gimsøy section between 50–650 m and 50–1500 m were identical to within 0.1 Sv, hence the limited vertical range of our transport calculation does not introduce additional errors in the baroclinic contribution.

Together with the temporal averages and one standard deviation, $\sigma$, we also report the standard error of the mean and a representative total transport error estimate. The standard error is calculated as se $= \sigma/\sqrt{n}$, using a degrees of freedom ($n$) taking into account the decorrelation time scale of 7 days (Sect. 5). We calculate a representative transport error estimate, for
winter, summer and annual data points separately, accounting for the time variability in statistics. At each mooring, we assume root-mean-squared errors of about 20% (4 km) in the effective width and 0.05 m s$^{-1}$ in depth averaged current (corresponding to 30 m$^2$ s$^{-1}$ of transport density). A simple calculation using these figures, ignoring the statistics, would lead to an error of 0.12 Sv. Using the mean and $\sigma$ of observed transport density (for winter, summer and all data separately), we generate 100 random data points from a normal distribution and calculate the transport (without imposed error) using 20.1 km width. The
distribution of transport is approximately normal in each season, and this assumed distribution for error analysis is justified.

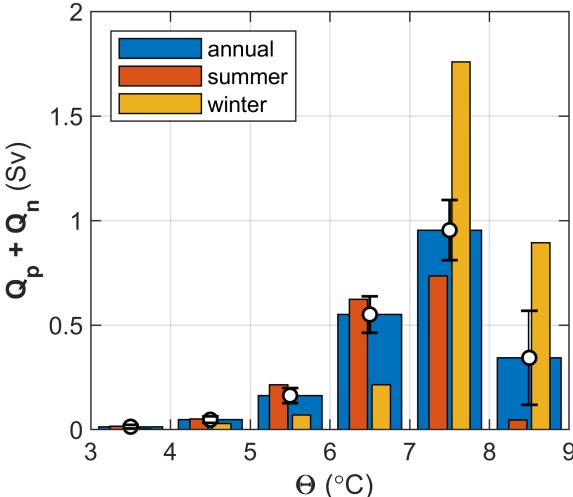

**Figure 7.** Total net transport ($Q_p + Q_n$) in the 50–650 m depth range, averaged in temperature classes for the entire record (annual), summer (JJA) and winter (DJF) months. Error bars (± standard error) are shown for the annual averages.

We then generate 100 values for transport density and width from a random distribution with imposed errors, and calculate the total transport (with error). The root-mean-squared value of the difference between transport values with and without error from this 100-point realization gives one error estimate. We draw 1000 bootstrap error estimates and average them to obtain the reported error. The transport error is 0.8 Sv for the annual average, 0.7 Sv for summer and 0.9 Sv for winter averages. This is typically less than the standard deviation and 3-4 times the standard error (Table 2).

**Table 2.** Volume transport calculations. Positive transport, $Q_p$ is directed northwest out of section, $Q_n$ is southeastward, and $Q$ is the total AW transport with $\Theta \geq 5°C$. $n$ is the degrees of freedom (daily data points divided by decorrelation time of 7 days). The values in square brackets are [$\pm\sigma$; $\pm$se], where $\sigma$ is the standard deviation, and se is the standard error (se $= \sigma/\sqrt{n}$). Additionally a total error estimate for $Q$ (see text) is given.

| Period | n | Transport (Sv) | | |
|---|---|---|---|---|
| | | $Q_p$ | $Q_n$ | $Q$ |
| Annual | 66 | 2.1 [±1.3; ±0.2] | -0.1 [±0.1; ±0.0] | 2.0±0.8 [±1.3; ±0.2] |
| Summer | 26 | 1.7 [±1.0; ±0.2] | -0.1 [±0.1; ±0.0] | 1.6±0.7 [±0.9; ±0.2] |
| Summer-16 | 13 | 1.5 [±0.8; ±0.2] | -0.1 [±0.1; ±0.0] | 1.4±0.6 [±0.7; ±0.2] |
| Summer-17 | 13 | 2.0 [±1.0; ±0.3] | -0.1 [±0.1; ±0.0] | 1.9±0.7 [±0.9; ±0.3] |
| Winter | 13 | 3.0 [±1.9; ±0.5] | -0.0 [±0.1; ±0.0] | 2.9±0.9 [±1.9; ±0.5] |

There is large variability in $Q$ with 1 to 4 Sv oscillations at 2 to 4 weeks time scale (Fig. 6a). The transport variability can be due to the current meandering outside the moorings, rather than a change in the along-slope transport. Transport maxima were observed in winter. The transport approached zero at the trough of the oscillations, but the flow reversal was negligible. Total AW transport was typically northward. Monthly averaged transport of AW increased three-fold in fall and winter with a monthly-average maximum of about 3.6 Sv in December, from about $1 - 2$ Sv in summer (Fig. 6b). The transport in temperature classes is shown in Fig. 7. When averaged over summer and winter months, separately, transport in high temperature classes (7–9°C) was stronger in winter whilst the low temperature classes (4–7°C) were stronger in summer. This is because the maximum AW-layer averaged temperatures occurred in winter (e.g., compare the winter and summer temperature profiles at MW, Fig. 4a), when the transport was also large (Fig. 6). In winter, the vertical mixing of warm surface layer resulted in a low stratified AW layer of 7–8°C. The largest transport was in the 7–8°C water for both seasons. We hypothesise that the largest warm water transport in winter is a consequence of the annual cycle of depth-averaged temperature coinciding with the time of strongest barotropic currents in winter. A seasonal variability with transport and temperature maxima in winter and minima in the autumn was also observed in the Svinøy section with an annual cycle amplitude in currents of about $0.10 \, \mathrm{m \, s^{-1}}$ (Orvik and Skagseth, 2005).

Statistics of the volume transport are summarized in Table 2. Overall, AW transport averaged over the entire record was 2.0±0.8 Sv (± total error; $\sigma = 1.3$ Sv, se = 0.2 Sv). Winter average (2.9±0.9 Sv) was larger than the summer average (1.6±0.7 Sv), significant when considering the se, but not the total error. Averaged separately, transport was stronger in summer 2017 relative to summer 2016, increasing from $1.4 \pm 0.6$ Sv to $1.9 \pm 0.7$ Sv in summer 2017. The difference was significant relative to the se.

The crude estimate of the width of the slope current must be treated with caution. The sensitivity to the choice of mooring width is approximately linear. Reducing the total effective width by a factor of two, to 10 km, reduces the mean AW transport from 2.0 to 1.0 Sv. AW transports, on the other hand, are not sensitive to the definition of the AW temperature and vertical integration limits. Recalculating the transport using water with $\Theta \geq 3$ (instead of $\Theta \geq 5$) increases $Q$ by less than 0.1 Sv. While the upper layers are characterized by lower salinity water, the proportion of AW entrained into the upper 50 m should ideally be accounted for in the AW transport estimates. In the core of the slope current between MN and MW, salinity from the hydrographical atlas vertically averaged in the upper 50 m varies between 35.25 and 34.95 $\mathrm{g \, kg^{-1}}$ (not shown). Assuming shelf waters of salinity less than $34 \, \mathrm{g \, kg^{-1}}$, the fraction of AW in the mixed water would exceed 65% to 80%. We limit our estimates at 50 m mainly because of lack of reliable current measurements. Including the upper 50 m by extending the uppermost available current measurement to the surface and assuming 100% AW fraction, increases the total mean transport by 0.3 Sv (from 2.0 Sv), well within the error estimates.

# 7 Climatological structure and comparison with the Svinøy section

There is a substantial transformation of AW between the Svinøy (63°N) and Gimsøy (69°N) sections, discussed in detail by Bosse et al. (2018). Analyses on temperature/salinity space and in isopycnal layers showed that AW was progressively

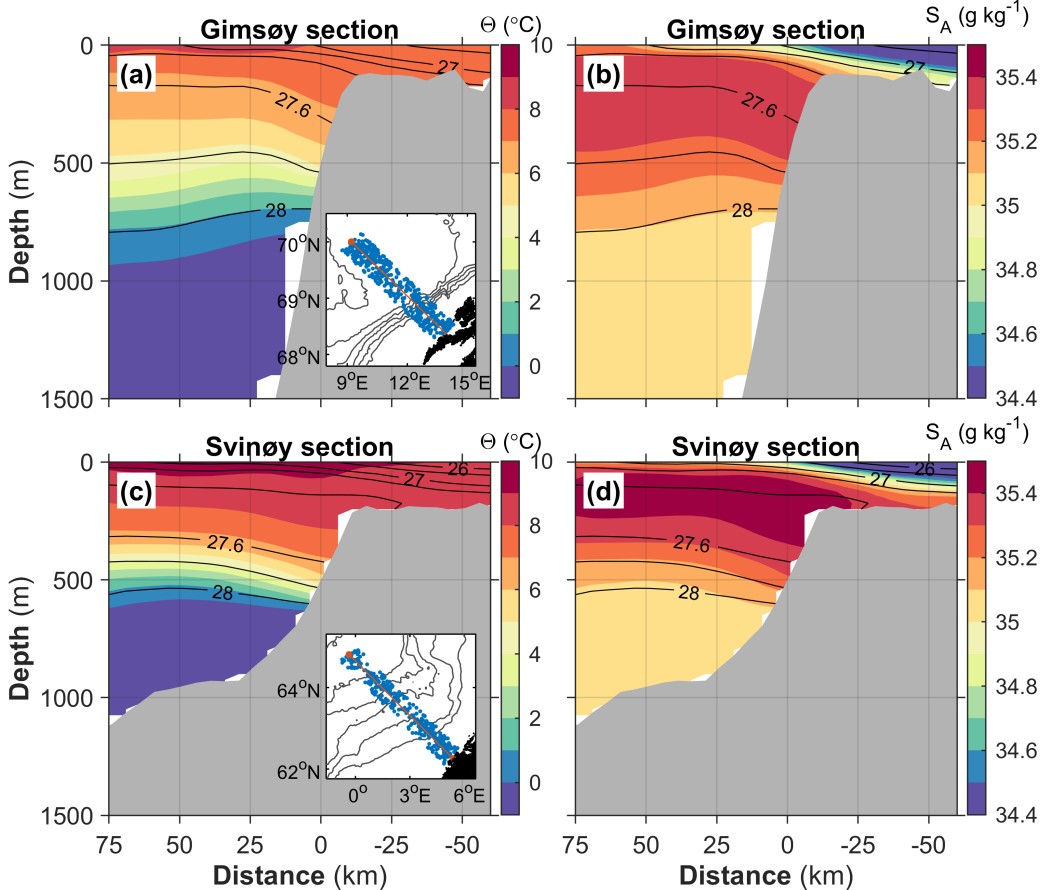

**Figure 8.** Mean (a-c) $\Theta$ and (b-d) $S_A$ distribution along the Gimsøy and Svinøy sections obtained from the Nordic Seas data set (Bosse and Fer, 2018). Contour interval is 1°C for $\Theta$ and 0.1 g kg$^{-1}$ for $S_A$. Salinity is saturated at 34.4 g kg$^{-1}$, but minimum values are 33.0 at Svinøy and 33.9 at Gimsøy. Isopycnals (potential density anomaly referenced to surface pressure, $\sigma_0$, black) are drawn at 0.2 kg m$^{-3}$) interval to 27 kg m$^{-3}$, followed by 26.5 and 26 kg m$^{-3}$ for shelf waters. An inset map for each section shows the profiles used, located within 25 km distance from the sections. Distance is referenced to the 500 m isobath.

transformed to denser isopycnals. While the most important transformation occurred in the western part of the Lofoten Basin,
lateral exchanges generated by instabilities of the slope current substantially modified the characteristics of the AW transported from the Svinøy to Gimsøy section. A climatological view of the hydrography in the Svinøy and Gimsøy sections shows the important cooling and freshening of AW (Fig. 8). As the AW is modified, isopycnals with potential density anomaly $\sigma_0$ less than 27.7 kg m$^{-3}$ rise. At the core of the slope current, the displacement of the 27.5 isopycnal reaches 150 m, switching from being located below the AW core to above. This is also where the largest spiciness injection - an indicator of water mass
transformation by diapycnal mixing - by vertical mixing was reported (Bosse et al., 2018). Deeper isopycnals sink from Svinøy to Gimsøy, which could be related to the intermediate waters subducted along the Mohn Ridge front and AW transformations

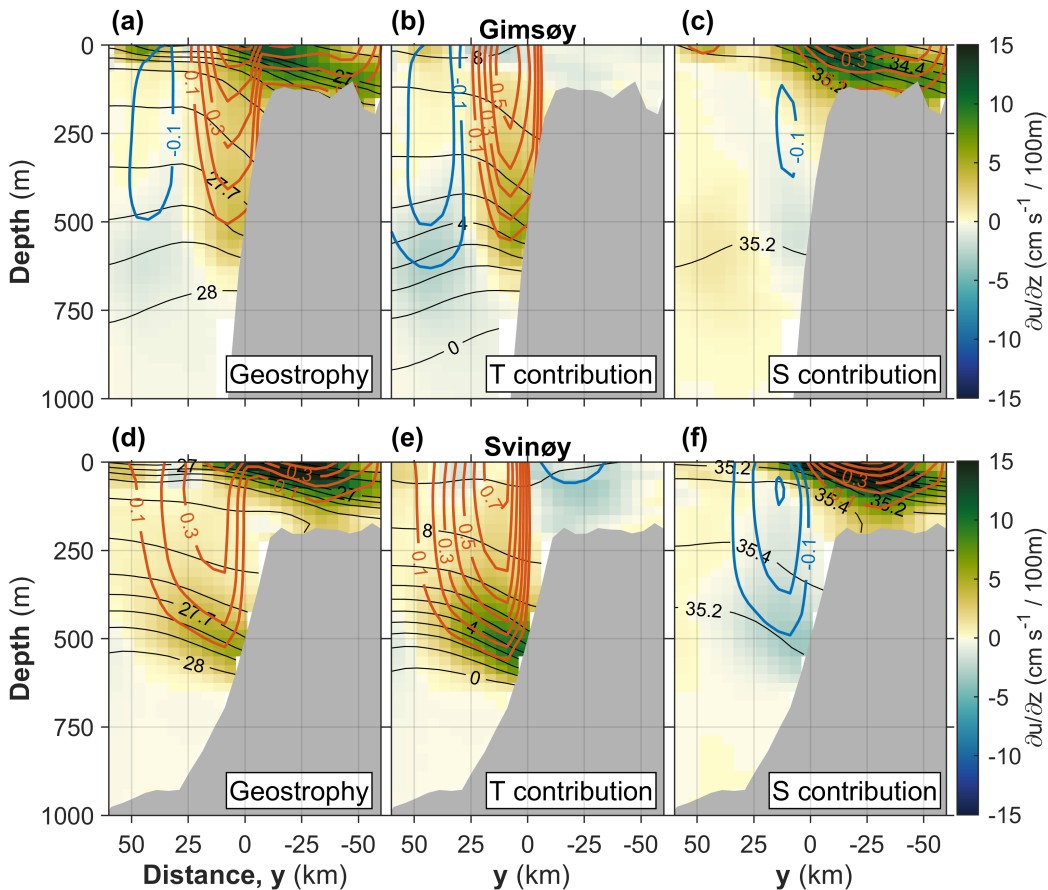

**Figure 9.** Vertical shear from thermal wind balance for (upper row) Gimsøy and (lower row) Svinøy sections using the annual-mean hydrography shown in Fig. 8. Panels (a) and (d) are the total geostrophic shear, (b) and (e) are the thermal contribution, and (c) and (f) are the haline contribution to shear. Vertically-integrated shear is also contoured (blue: negative; red: positive values). Distance is referenced to the 500 m isobath. Isolines are drawn at 0.1 kg m$^{-3}$ for $\sigma_0$ (down to 27 kg m$^{-3}$, and with additional 26.5 and 26 kg m$^{-3}$ contours), 1°C for $\Theta$, 0.2 g kg$^{-1}$ for $S_A$, and 0.1 m s$^{-1}$ for vertically-integrated shear.

in the Lofoten Basin, decreasing the stratification in the AW pycnocline. As a result of winter mixing driven by intense air-sea fluxes, the AW pycnocline in the Lofoten Basin is more diffuse and deeper at around 800 m (vs. 500 m farther south). The cross-slope temperature and salinity gradients across the slope current also exhibit a different structure, suggesting different contributions to geostrophic currents (via thermal wind balance) and a change with latitude in baroclinicity of the slope current (Fig. 9).

The cross-slope gradients are relatively weaker at Gimsøy compared to Svinøy section, and so are the temperature contribution (positive at the slope, negative on the shelf) and haline contribution (negative at the slope, positive on the shelf) to the geostrophic shear (Fig. 9). Furthermore, the coastal current core - identified by the positive shear driven by salinity on the

320 shelf - interacts more strongly with the slope current at Gimsøy. This can be explained by the steeper slope of the Lofoten Escarpment, which has a stronger control on the mean position of the slope current. Note that the broader region of isopycnal gradients at Svinøy does not necessarily imply a broader current, but could result from a more variable position linked to a weaker topographical control by the steepness of the slope.

To further compare the baroclinicity of the slope current at these two locations, we vertically integrated the different contribu-
325 tions to the geostrophic shear with a level of no motion at the bottom (geostrophic velocity contours in Fig.9). The baroclinicity of the slope current indeed increases with latitude: poleward geostrophic currents exceed $0.6$ m s$^{-1}$ at Gimsøy compared to about $0.4$ m s$^{-1}$ at Svinøy, despite the stronger contribution from vertically-integrated shear due to temperature ($0.75$ m s$^{-1}$ at Svinøy vs. $0.56$ m s$^{-1}$ at Gimsøy). A strong negative shear due to salinity counter-balances the thermally-driven geostrophic shear of the current at Svinøy (reaching $-0.31$ m s$^{-1}$ integrated from bottom to 150 m, $-0.25$ m s$^{-1}$ to the surface). At Gimsøy,
this value reaches only $-0.12$ m s$^{-1}$ from the bottom to 250 m, and becomes insignificant when integrated to the surface. This suggests that the cross-slope salinity gradient is important for the baroclinicity of the slope current, even in a region where temperature accounts for most of the density variations. Changes in the baroclinicity of the slope current can thus be expected following the recent AW freshening observed in the Nordic Seas (Mork et al., 2019).

## 8 Energetics

The kinetic energy content and variability of the slope current, and conversion rates associated with barotropic and baroclinic instability of the current are presently unconstrained by observations. Using our limited mooring records, we attempt to quantify the energetics of the slope current at the Lofoten Escarpment. For the following analysis, we obtained the fluctuations, denoted by primes, by band-pass filtering the hourly data with cutoff frequencies corresponding to 14 day and 35 hours.

We start with the variability in depth-averaged along and cross slope currents, the horizontal eddy kinetic energy density,
EKE, and their relation to wind forcing. The EKE in units of J kg$^{-1}$ or m$^2$ s$^{-2}$, is

$$\text{EKE} = \frac{1}{2}\left(u'^2 + v'^2\right) . \tag{1}$$

The along-slope current variability and the evolution of EKE were partly forced by the along-slope wind modulating the geostrophic shear by cross-front Ekman transport (Fig. 10a-c). The annual average wind speed was $4.3 \pm 2.3$ m s$^{-1}$ (in this section the figures after $\pm$ are 1 $\sigma$). In winter, the average and maximum wind speed values were 6 m s$^{-1}$ and 11 m s$^{-1}$,
respectively. Depth averaged (200–600 m) $u$ at MW was $0.15 \pm 0.12$ m s$^{-1}$ with a maximum of 0.6 m s$^{-1}$ in winter (winter average was $0.25 \pm 0.16$ m s$^{-1}$). The maximum correlation between depth-averaged $u$ and the along-slope component of the wind $W_x$ was obtained at 2 day lag ($r$=0.6). While no significant correlation was detected with the cross-slope component, $v$ increased in amplitude and variability in winter (from its annual average of $0.03 \pm 0.04$ m s$^{-1}$) to $0.05 \pm 0.07$ m s$^{-1}$ reaching a maximum of $0.26$ m s$^{-1}$.
From 30-day moving averages, EKE was $(65 \pm 38) \times 10^{-4}$ m$^2$ s$^{-2}$ with a maximum of $185 \times 10^{-4}$ m$^2$ s$^{-2}$. Daily average values were similar, but with 3 times larger standard deviation and a maximum of $790 \times 10^{-4}$ m$^2$ s$^{-2}$. The maximum EKE

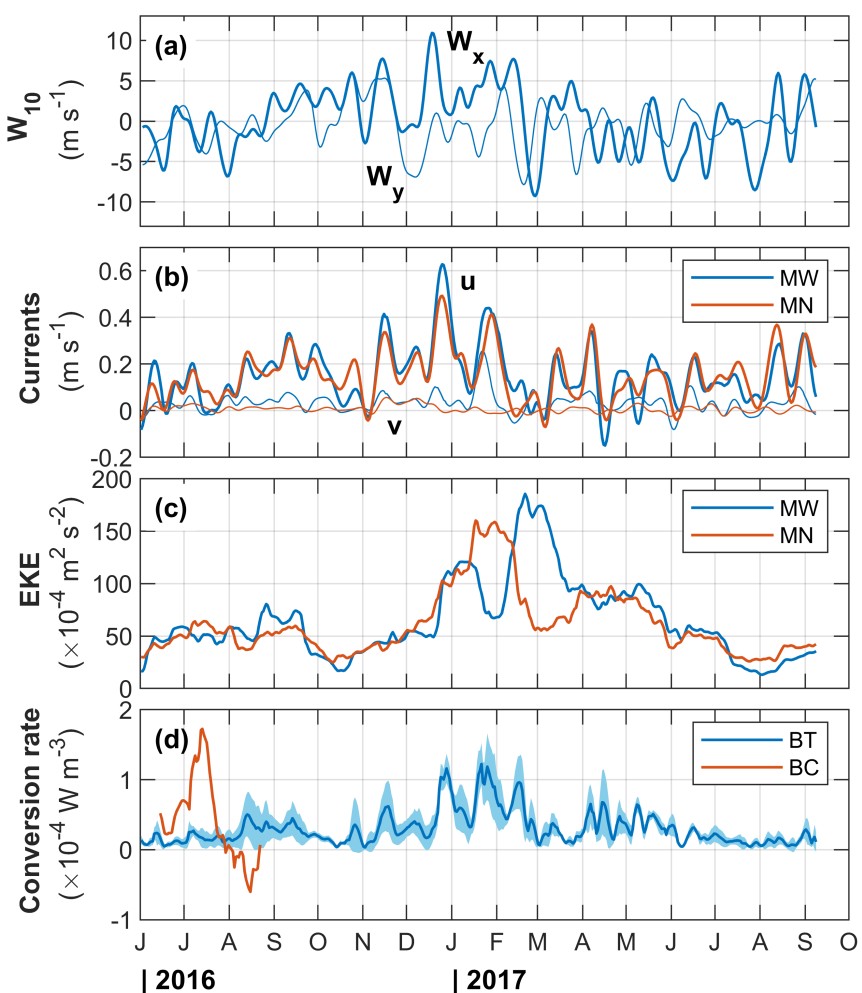

**Figure 10.** Time series of a) ERA5 wind along-slope ($W_x$) and cross-slope ($W_y$) components, b) 200–600 m averaged $u$ and $v$ measured at mooring MW (blue) and MN (red), c) 200–600 m averaged EKE at MW (blue) and MN (red) and d) barotropic (BT, blue) and baroclinic (BC, red) conversion rates. BC is at 400 m level and only available for 3 months. BT is the depth-average and one standard deviation envelope over calculations at 200, 300, 400, 500 and 600 m. All curves are 30-day moving averages.

was observed in winter, consistent with stronger and favorable downfront winds. When averaged over winter months EKE was $(100 \pm 41) \times 10^{-4}$ m$^2$ s$^{-2}$.

An estimate of the baroclinic (BC) and barotropic (BT) conversion rates can be made assuming no variability in the along-
slope direction and the cross-slope gradients dominate. Similar calculations were made both in idealized (channel) model studies (e.g., Spall et al., 2008) and using mooring array data in the West Spitsbergen Current (von Appen et al., 2016), in

the East Greenland Current (Håvik et al., 2017), and across the bondary current at Beaufort shelf break and slope (Spall et al., 2008). A positive value of BC indicates conversion from mean potential energy into EKE, by growing eddies extracting energy from the mean state. The conversion from mean kinetic energy into EKE is quantified by BT. In this case, the kinetic energy is extracted from the mean flow by eddies transporting along-slope momentum down the mean velocity gradient (e.g., Spall et al., 2008). The baroclinic conversion rate can be approximated by

$$\mathrm{BC} = g\overline{v'\rho'}\frac{\partial z}{\partial y} \ , \tag{2}$$

where the cross-slope velocity fluctuation $v'$, and the density fluctuation $\rho'$ are obtained by 14-day – 35-h band-pass filtering the hourly data, $\rho_0 = 1027\,\mathrm{kg\,m^{-3}}$ is a reference density, $g$ is gravitational acceleration, and we applied a temporal averaging (overbar) using 30 day moving averaging. The mean isopycnal slope, $\partial z/\partial y$, was calculated as $(\partial\overline{\rho}/\partial y)/(\partial\overline{\rho}/\partial z)$. The barotropic conversion rate can be approximated by

$$\mathrm{BT} = -\rho_0\overline{u'v'}\frac{\partial \overline{u}}{\partial y} \ . \tag{3}$$

As in the BC calculations, fluctuations are the 14-day – 35-h band-passed hourly data, and time averaging is over 30 days. While the velocity data coverage is good in both moorings, density (through salinity measurement) measurements are limited. At MW, density measurements are available at target depths of 75, 380, 980 and 1476 m. At MN, the near-bottom sensor (648 m) recorded throughout, but the sensors at 165 and 455 m recorded only until September (the water column line was cut 3 months after the deployment). Note that motion-corrected mooring data were gridded and interpolated. Based on the density measurement coverage, we picked the 400-m level as a representative depth (in AW and in the wedge of AW current with steep isopycnals, Fig. 2) where we can obtain vertical and lateral gradients, but only for 3 months into the record. We calculated the vertical gradient at 400 m at MW using the records at 300 and 500 m, and the lateral gradient from the records at 400 m. Whilst the baroclinic conversion rate time series is limited only to three months, the barotropic conversion rate can be calculated for the entire duration. We computed BT at 300, 400, 500 and 600 m depths. Results are summarized in Fig. 10d.

Average barotropic conversion rate (averaged over both moorings, over multiple levels and over 14 months) was $(0.3\pm0.2)\times 10^{-4}\,\mathrm{W\,m^{-3}}$. Maximum value reached $1.2\times10^{-4}\,\mathrm{W\,m^{-3}}$. The baroclinic conversion rate (only available in the summer for the first three months of the mooring period) was comparable, $(0.4\pm0.6)\times10^{-4}$ with a maximum of $1.7\times10^{-4}\,\mathrm{W\,m^{-3}}$. For reference, a conversion rate of $10^{-4}\,\mathrm{W\,m^{-3}}$ for 1 day accounts for $\rho_0$ EKE of $O(10)\,\mathrm{J\,m^{-3}}$, or EKE of $O(100)\times10^{-4}\,\mathrm{m^2\,s^{-2}}$.

Observed EKE and the conversion rates at the Lofoten Escarpment can be compared to other relevant observations. In Fram Strait von Appen et al. (2016) analyzed 12 year long time series from moorings with focus on the West Spitsbergen Current. EKE at 75 m depth was $50\times10^{-4}\,\mathrm{m^2\,s^{-2}}$ in summer, and increased to $200\times10^{-4}\,\mathrm{m^2\,s^{-2}}$ in winter. At 250 m depth the magnitude was approximately reduced to half. These values, overall, are similar to the EKE at the Lofoten slope. In terms of baroclinic and barotropic conversion rates, the two sites are also comparable: in the West Spitsbergen Current BT was on the order $0.1\times10^{-4}\,\mathrm{W\,m^{-3}}$, and BC at 75 m was $0.5\times10^{-4}\,\mathrm{W\,m^{-3}}$ in summer, increasing to $1.5\times10^{-4}\,\mathrm{W\,m^{-3}}$ in winter.

Summer mean and maximum values are identical (within measurement uncertainties) to the corresponding values from our observations at 400 m in summer.

Using a mooring array Håvik et al. (2017) analyzed the structure and variability of the shelfbreak East Greenland Current, for the period September 2011 to August 2012. EKE at 100 m was up to $700 \times 10^{-4}$ m$^2$ s$^{-2}$ in November 2011 when a reversal of shelf break current was observed, otherwise typical values varied between $10 \times 10^{-4}$ and $100 \times 10^{-4}$ m$^2$ s$^{-2}$, similar to the values at the Lofoten Escarpment. Ignoring the energetic reversal event, BT at 100 m was on the order $0.1 \times 10^{-4}$ W m$^{-3}$ and BC varied in the range of $(1 - 5) \times 10^{-4}$ W m$^{-3}$; both conversion rates are similar to those in the West Spitsbergen Current and the slope current.

The conversion rates calculated from our moorings may not be representative of the volume-averaged conversion rates over the slope region. In order to assess this, we compared the observations to high-resolution numerical model results in Sect. 9.

## 9  Conversion rates from a high resolution model

In order to better interpret the conversion rates obtained from moorings, we calculate volume-averaged conversion rates in the region using the outputs from high-resolution ROMS runs (Sect. 2.2). We first compute the baroclinic and barotropic conversion rates over a domain covering the slope region identified in Fig. 11. The conversion rates in a 3D, right-handed coordinate system are formulated in Olbers et al. (2012, , pp.376-377). The baroclinic conversion rates are computed from

$$\text{BC} = -\rho_0 \overline{\mathbf{u}'b'} \cdot \nabla \overline{b}/N^2 = g \left( \frac{\partial \overline{\rho}}{\partial z} \right)^{-1} \left( \overline{u'\rho'} \frac{\partial \overline{\rho}}{\partial x} + \overline{v'\rho'} \frac{\partial \overline{\rho}}{\partial y} \right). \tag{4}$$

Here $\mathbf{u} = (u, v)$ is the horizontal velocity field (formulation is valid for both the model grid and the along/across-slope rotated coordinate system), $b = -g\rho/\rho_0$ is the buoyancy, $N^2 = -\frac{g}{\rho_0} \frac{\partial \overline{\rho}}{\partial z}$ is the buoyancy frequency, $\rho$ the potential density referenced to surface, $g$ the gravitational acceleration, and $\rho_0 = 1027$ kg m$^{-3}$ is a reference density. The primes denote deviations from an average state (overbar), averaged over multiple eddy time scales, e.g. for velocity $u' = u - \overline{u}$. A positive value of BC indicates a transfer of potential energy from the mean flow to eddies.

We calculate the barotropic conversion rates from

$$\text{BT} = -\rho_0 \left( \overline{u'\mathbf{u}'} \cdot \nabla \overline{u} + \overline{v'\mathbf{u}'} \cdot \nabla \overline{v} \right) = -\rho_0 \left( \overline{u'u'} \frac{\partial \overline{u}}{\partial x} + \overline{u'v'} \left( \frac{\partial \overline{u}}{\partial y} + \frac{\partial \overline{v}}{\partial x} \right) + \overline{v'v'} \frac{\partial \overline{v}}{\partial y} \right). \tag{5}$$

A positive value of BT indicates a transfer of kinetic energy from the mean flow to eddies.

We compute BC and BT after interpolating the model fields to uniform $z$-levels of 10 m vertical spacing. The time averaging and fluctuations are calculated over monthly windows to avoid any seasonal bias. We arbitrarily chose the year 1999 from the model fields (available from 1996 to the end of 1999). Monthly conversion rates are then averaged vertically between 100-1000 m depth (i.e., we exclude the near-surface variability). A global annual average is then obtained by averaging over these 12 months. Results are shown in Fig. 11.

The baroclinic conversion rates are typically positive and largest along the slope, indicating that potential energy is extracted from the slope current to feed eddies that are generated there. The barotropic conversion rates, on the other hand, show larger

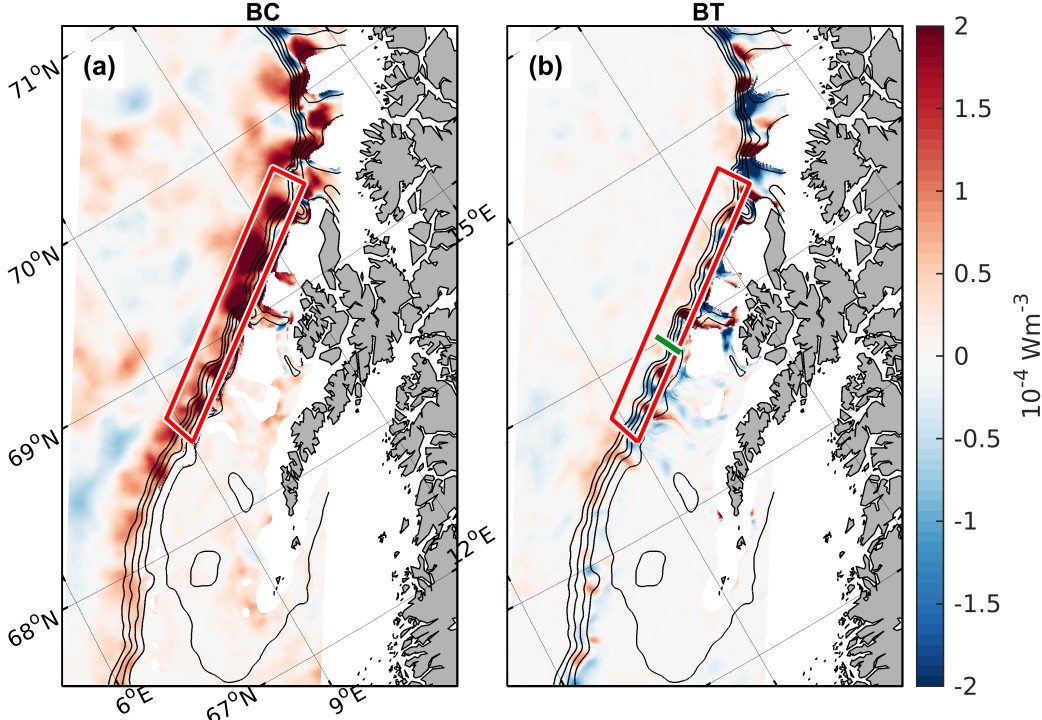

**Figure 11.** Maps of a) baroclinic and b) barotropic conversion rates averaged over one year (1999) and between 100-1000 m depth or to bottom in shallower areas. Longitudes and latitudes are identical in both panels and are only labelled in (a). The red box is the slope region where volume-averaged conversion rates shown in Fig 12a are computed. The green line across the slope in (b) marks a segment across the mooring positions used for comparison of volume averaged and segment/virtual mooring calculations (see text and Fig 12b). Black contours show the 200, 400, 600, 800 and 1000 m isobaths.

spatial variability. The magnitudes are smaller and the sign often changes. The baroclinic processes therefore appear to be the main contributor to the conversion of energy from the mean flow to eddies along the slope.

Monthly conversion rates over the slope, volume averaged over the red box identified in Fig 11a and between 100 and 1000 m depth show that the baroclinic conversion rates dominate (Fig 12a), implying the baroclinic instability of the slope current extracts energy from the mean flow to eddies.

The motivation here is to assess whether the conversion rates obtained from a mooring array are representative of the volume-averaged values. To do this we define a segment across the slope (green in Fig 11b), that stretches between the mooring positions of MW and MN and extend further by 10 km at both sides. We then perform two types of calculations: 1) we compute BT and BC at the model grid resolution and average along the entire segment, and 2) we compute BT and BC using model data from the virtual mooring positions. To be consistent with the observations we apply Eqs. 2-3, rotate the coordinate system to along and across isobaths, calculate BC only at 400 m depth, and vertically average BT between 200 and 600 m. The motivation of

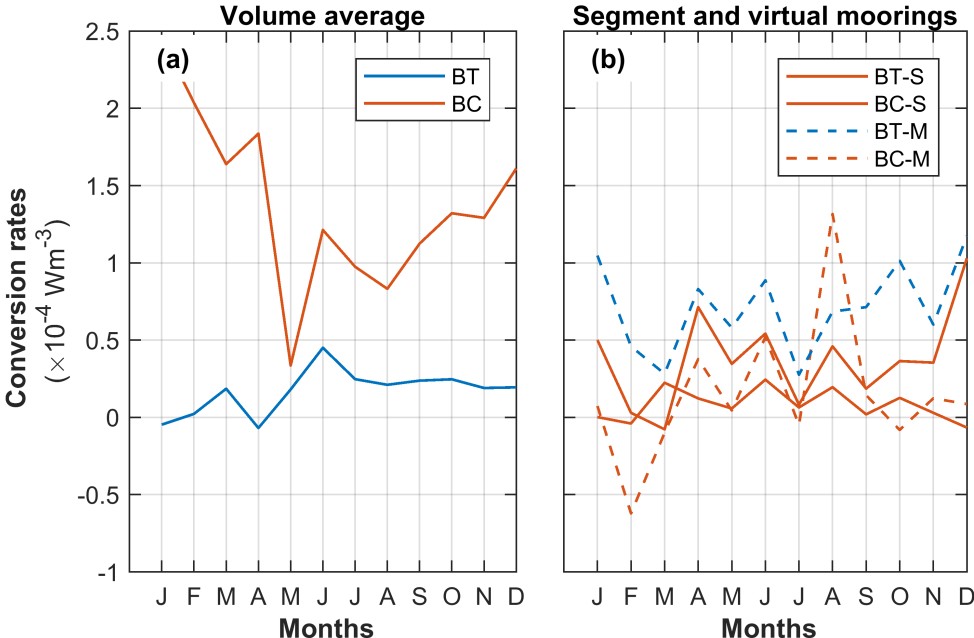

**Figure 12.** Monthly-averaged barotropic (blue) and baroclinic (red) conversion rates a) vertically averaged between 100 and 1000 m depth and inside the red box in Fig 11 and b) along a segment across the slope (green line in Fig 11b) using model horizontal resolution (BT-S and BC-S, solid lines) and using two virtual moorings (BT-M and BC-M, dashed lines). Calculations were made using Eqs. 4-5 in (a), and using Eqs. 2-3 in (b). The baroclinic conversion rates in (b) are shown at 400 m depth, whereas the barotropic conversion rates are averaged between 200 and 600 m depth, i.e., directly comparable to the observations.

performing the segment calculation is to better resolve the lateral shear (based on about 40 grid points compared to only two virtual moorings). The conversion rates are shown in Fig 12b.

While there are differences between the segment and virtual mooring estimates (Fig 12b), the conversion rates are comparable with no systematic differences. Lateral shear and isopycnal slopes using only two moorings separated by about 5 km could thus be used in calculations of the conversion rates in one transect. We also note that BT is similar to the observations (blue

line in Fig. 10d) with magnitudes between $(0 - 1) \times 10^{-4}$ W m$^{-3}$ and maximum values around $1 \times 10^{-4}$ W m$^{-3}$. Observed BC is available only in the summer months (red line in Fig. 10d), and compare fairly well with the BC from virtual moorings. However, a comparison with the volume-averaged conversion rates shows that calculations using virtual moorings alone overestimate BT, underestimate BC, introduce spurious changes in sign, and are not representative of the overall conversion rates on the slope.

The discrepancy in BT is partly due to the different depth averaging (100 to 1000 m vs. 200 to 600 m, note the latter range is constrained by available observations whereas the former covers the depth range of interest on the slope region, excluding the upper surface processes), and partly because the volume-averaged calculations include the divergent terms (first and last

term in Eq. 5) in addition to the terms related to shear (second term). The highly variable spatial structure observed in BT cannot be resolved with a high-resolution single segment or a couple of moorings. Furthermore, volume averaging over BT with changing signs leads to a negligible average BT, which cannot be resolved with the moorings. The discrepancy in BC is mainly because the volume-averaged calculations are based on a depth average between 100 and 1000 m, whereas the mooring calculations are only taken at 400 m depth due to limited observations. BC cannot be captured by the calculations from a single level.

Based on the analysis of the model outputs, we conclude that the mooring-derived conversion rates must be interpreted with caution and may not be representative of the real conversion rates in the region. While we cannot confirm using the limited observations, the model results suggest that the average conversion rates on the Lofoten Escarpment are likely dominated by baroclinic instability of the slope current.

## 10    Summary and Conclusions

The Norwegian Atlantic Slope Current at the Lofoten Escarpment is described using 14-month long mooring records in the period from June 2016 to September 2017. Despite the limited number of moorings, the observations resolve the core of the current from 200 to 650 m depth over the shelf break and the upper continental slope. The data set is the first moored observations on yearly time scale from this region, and offers important constraints on mean properties, transport rates, temporal variability, and energy conversion rates of the slope current.

The 200–600 m averaged current shows an annual cycle with amplitude of 0.1 m s$^{-1}$ with strongest currents in winter, and has a temporal average of 0.15 m s$^{-1}$. 14-day low pass filtered along-slope currents reach 0.8 m s$^{-1}$, lasting for 1 to 2 weeks and extend as deep as 600 m. The variability in the along-slope current is partly forced by the along-slope wind stress, with a maximum correlation of 0.6 at 2 day lag. In contrast to observations in Svinøy, the slope current is not barotropic and varies strongly with depth (shear of 0.05 to 0.1 m s$^{-1}$ per 100 m in all seasons).

The average volume transport of Atlantic Water is 2.0±0.8 Sv, with summer and winter averages of 1.6±0.7 and 2.9± 0.9 Sv, respectively. The largest transport is associated with warm water in all seasons, and the water temperatures are the highest in winter.

Calculations of the barotropic and baroclinic conversion rates from the moorings are supplemented by results from a high resolution numerical model. While the conversion from mean kinetic energy into eddy energy (e.g. barotropic instability) is likely negligible over the Lofoten Escarpment, the baroclinic conversion from mean potential energy into eddy kinetic energy (e.g. baroclinic instability), can be substantial with volume-averaged values on the order of $10^{-4}$ W m$^{-3}$. Eddy kinetic energy and conversion rates in the slope current are comparable to the published results from the West Spitsbergen Current and the East Greenland Current.

Fishing activity in the region makes it highly challenging to maintain moorings; however, extended time series with better cross-slope and vertical coverage are needed to study the dynamics and variability of the slope current. The attempts to calculate (observation-based) energy conversion rates remain inconclusive. Utilization of autonomous underwater vehicles, such as

gliders, can help collecting high quality observations, but will be difficult to operate in the strong boundary current. The slope current and its instability is an important player in the energetics of the Lofoten Basin and merits further studies.

*Data availability.* Mooring data used in this analysis are available from Fer (2020), from https://doi.org/10.21335/NMDC-1664980441 through the Norwegian Marine Data Centre with Creative Commons Attribution 4.0 International License. The data set of the Nordic Seas (Bosse and Fer, 2018) is available from https://doi.org/10.21335/NMDC-1131411242, through the Norwegian Marine Data Centre with Creative Commons Attribution 4.0 International License. Other environmental data are obtained using Copernicus Climate Change Service (C3S) (2017), and European Centre for Medium-range Weather Forecast (ECMWF) (2011). Sea level anomaly data are obtained from the E.U. Copernicus Marine Service Information, product SEALEVEL_GLO_PHY_L4_REP_OBSERVATIONS_008_047.

*Author contributions.* I.F. conceived the experiment, processed and analyzed the mooring data and wrote the paper. A.B. extracted and analyzed the surface forcing and the Nordic Seas climatology data. J.D. calculated conversion rates from existing model fields. All authors contributed to interpret and discuss the results and reviewed the manuscript.

*Competing interests.* I. Fer is a member of the editorial board of Ocean Science, but other than that the authors declare no competing interests.

*Acknowledgements.* This study received funding from the Research Council of Norway, through the project *Water mass transformation processes and vortex dynamics in the Lofoten Basin in the Norwegian Sea (PROVOLO)*, project 250784. The ROMS simulation was made by Marta Trodahl and Nils M. Kristensen of the Norwegian Meteorological Institute and run on resources provided by UNINETT Sigma2-The National Infrastructure for High Performance Computing and Data Storage in Norway. We thank the crew and participants of the deployment and recovery cruises, and particularly Helge Bryhni, Algot Peterson and Henrik Søiland for their help with the mooring work. We thank Kjell Arild Orvik for making available the Svinøy mooring data used in the discussion forum. Insightful and constructive comments from Michael A. Spall and two anonymous reviewers helped improve the discussion version of this manuscript.

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
