# Peer review of "Norwegian Atlantic Slope Current along the Lofoten Escarpment"

_Ocean Science, 2020_

## Short Comment (SC1) · 21 Mar 2020

Unfortunately my personal note (line 243-244) went unnoticed when submitting the final version of our Discussion paper. I apologize for the confusion this may have caused. For clarification, we used an averaging time scale of 30 days (not 14 days). This ensures sufficient averaging over multiple eddy time scales and is also consistent with the calculations from the model fields (Appendix A). Thanks, Ilker
* * *

---

## Referee Comment (RC1) · Anonymous Referee #1 · 25 Mar 2020

Review of the manuscript: "Norwegian Atlantic Slope Current along the Lofoten Escarpment" by Ilker Fer et al. (os-2020-15).

This work is based on a comprehensive data set, addresses important questions, and is generally well written and illustrated. Thus, I am confident that it deserves publication in something close to its present form, but it does contain some confusing aspects and details that need to be clarified before final acceptance, as elaborated below.

Main comments

1) Velocity components: Figure 1b shows an (x,y) coordinate system and you expect velocity components to follow the standard notation (u,v). This is confirmed on lines 97-98: "Current components are along-slope, u, and across-slope, v", but then this

statement is followed by: "(In the figures, we explicitly use the notation ua and ux, respectively.)" without explaining, which is which (and ux is not along the x-axis). Most of the manuscript seems to keep the (ua, ux) notation, but Sect. 7 with Eqs. (2) and (3) partly returns to the (u,v) notation. This is unnecessarily confusing. Stick to one notation. I would suggest (u,v).

2) Projected distance: On lines 181-182, you write: "The moorings are separated by approximately 6 km (horizontal distance between the locations), and when projected onto the cross-slope section to their respective isobaths, the distance is about 8 km". How can a projected distance become larger than the distance ? Using the positions in Table 1, I get a distance of 5.3 km, which projected onto the cross-slope direction ought to be around 4 km. This projected distance is used extensively in the manuscript. I am not sure that any main results are substantially affected by this, but again it is unnecessarily confusing and has to be corrected.

3) The width of the current for transport calculation: As I read your description (lines 182-187), your transport calculation is equivalent to multiplying the average transport density from the two moorings by 28 km (which according to the previous comment ought to be 24 km), but you do not mention what you do with the shallowing bottom in the 10 km inside of MN (to 250 m according to line 184). The statement on lines 184-185: "hence assign a 14 km effective width of water column to each mooring" does not indicate that this was taken into account. This should either be briefly clarified or the transport corrected if it has not been taken into account. As for many similar studies, the width of the current is probably the most uncertain aspect of the transport calculations, so is this uncertainty included in the uncertainties cited in Table 2 or are they just statistical (standard error ?). This should be clarified in the table heading and perhaps also in the text.

4) Transport of the top 50 m layer: The average volume transport cited in the abstract (2.8 Sv) seems not to include the top 50 m (line 213). I assume the reason to be that this layer is less saline than 35.17 on average (Figure 2) due to some admixture of

water from the Norwegian Coastal Current, but isn't the fraction of Atlantic water in this layer still » 50% ? Using S=35.17 as a lower boundary for Atlantic water does not necessarily imply that you should use the same criterion for an upper boundary. If you want to retain this, it should in any case be better justified in the text.

Details

5) line 7: "volume transport" -> "volume transport of Atlantic water"

6) line 19: "Iceland-Faroe ... channels" is not standard. You could use "gaps" instead of "channels"

7) line 110: "Norwegian Sea Deep Water" -> "deeper water". The NSDW is usually reserved for the deepest component, which is separated from AW by intermediate water masses.

8) Figure 2, third line of caption: "typical value" -> "typical lower limit".

9) lines 118-119 : The last sentence seems to contradict the previous sentence.

) line 168: "the two moorings" -> "MS and MN".

10) lines 175 onwards : In line 175, the letter "Q" (with or without subscripts) is defined as "transport density (m**2/s). After that, the same letter is used for "transport" (m**3/s). This should be corrected.

11) Figure 8: Consistent with my main comment 1) above, I suggest that you use (Wx,Wy) instead of (Wa,Wx). Having Wx perpendicular to the x-axis is confusing. In the caption to this figure, it might also be emphasized that the velocities in Figure 8b (presumably) are low-passed (not band-passed, as might be assumed from line 220).

12) line 324: Isn't there a rho-0 too much in the last equation ?

13) line 330: Isn't a parenthesis missing in the middle equation ?

14) Figure A1b: In my printed version of the manuscript, I could not identify any "magenta line". On the screen, there is something that could be magenta, but it does not resemble a line.

15) Appendix A: The contents of the appendix seem rather crucial to some of the results and the last parts of the abstract. You might consider pulling it into the main manuscript as a separate section just before the conclusion.

––––––––––––––––––––––––––––––

---

## Referee Comment (RC2) · Michael A. Spall (Referee) · 26 Mar 2020

This is a timely analysis of mooring data within the Norwegian Atlantic Slope Current on the eastern side of the Lofoten Basin. This region has been identified as a source of eddy kinetic energy and offshore eddy heat flux, which is important for the basin-scale stratification and air-sea exchange. The analysis is fairly straightforward and I recommend that it be published subject to relatively minor revisions. There are a couple suggestions for additional analysis that, while not crucial, would provide more context for the results.

Would it be possible to compare the transport in density and depth with that at the Svinoy section? If the transport there is barotropic, and here it is baroclinic, that implies

that there has been some upwelling between these two stations, or a loss of transport in the deeper layers. If the transports are similar, can you tell if the isopycnals have risen or if there has been a water mass transformation between these two sections? I think a more complete comparison with that upstream section can reveal more about what has happened between these locations. Even if the years are different, maybe you can consider the seasonal cycle, which should be representative.

Can the authors provide error bars for the velocity and transport estimates?

Introduction:

You might also reference Clark and Straneo (Observations of Water, Mass Transformation and Eddies in the Lofoten Basin of the Nordic Seas, JPO, 2015).

I had a 2010 paper in Ocean Modeling that would be more appropriate to reference than the 2010 JPO paper as it addresses the lateral eddy heat flux in (an idealized) Lofoten Basin (Spall,Non-local topographic influences on deep convection: An idealized model for the Nordic Seas, Ocean Modeling, 32, 72-85).

lines 120-124: It should be possible to quantify the source of the increased vertical shear, or at least break it down into temperature and haline contributions via thermal wind.

linbe 157: It might be useful to provide a scaling for the expected response to changes in the wind stress. One could calculate the onshore Ekman transport, downward deflection of the isopycnals, and the geostrophic response. The paper by Choboter et al. (2011, Exact Solutions of Wind-Driven Coastal Upwelling and Downwelling over Sloping Topography, JPO, 41, 1277-1295) provides analytic solutions but you might be able to do something useful just with simple scaling.

line 191: It seems likely that the transport variability is due to the current meandering outside the moorings (rather than a change in the along-slope transport), but this isn't explicitly mentioned.

Figure 7: I found this to be the most surprising part of the paper. Any ideas why there is more warm water in winter than in summer? When/where was this water last exposed to the atmosphere? Was this subducted in the previous summer? If you see the same phase at Svinoy, which is O(1000 km) upstream, that would argue against it simply being advected along the slope. I think some more discussion around this finding would be helpful. The penetration of AW down to 650 m depth is likely related to that being the sill depth upstream.

line 232: BC and BT were also calculated from a high resolution mooring array in Spall et al. (2008).

line 238-239: CHECK!NOT 1 MONTH?

lines 242-245: This justification is not very convincing, I suggest deleting it.

line 340: Magenta does not stand out compared to the colorbar, I suggest using a different color to mark the line.

Mike Spall mspall@whoi.edu

---

## Referee Comment (RC3) · Anonymous Referee #3 · 1 Apr 2020

Overview and general recommendation:

The manuscript describes the outcome of a mooring effort, carried out between Jun 2016 and August 2017 across the Norwegian Atlantic Slope Current off the Lofoten Islands at the so-called Lofoten Escarpment. The authors exploit the data from a mooring array that consisted of three deep sea moorings. Two of them, moorings MN and MW, were located about 6 km apart from each other across the slope current. A third mooring, MS was located almost 30 km further upstream close to the Grinvoy hydrographic repeat section. A fourth mooring, MB, was located in the interior of the Lofoten Basin and was not part of the analysis. The authors use the mooring records, mainly velocity data obtained from Longranger ADCPs as well as T/S information from MicroCATs or temperature loggers, to address the Atlantic Water (AW) layer within the Norwegian

Atlantic Slope Current that is captured by the moorings. While there are already descriptions of the Slope Current from the sections located upstream, the authors state that they provide the first mooring based description for the Grinvoy region off the Lofoten Islands. The authors describe the general nature of the velocity structure in the upper water column and find the strongest velocities in the winter period. This timing coincides with the time of the warmest temperatures observed in the AW layer. The authors furthermore infer transport time series for the two moorings MN and MW, explain their choice of a respective area over which the transport is calculated and finally quantify the volume transport for the AW layer. The authors furthermore address the forcing and find a correspondence between the along-stream wind forcing and the along-stream current component. Finally, the authors infer energy conversion rates from the mooring records, in particular baroclinic and barotropic conversion rates that describe the transfer of mean potential energy into eddy kinetic energy and the transfer of mean kinetic to eddy kinetic energy. The baroclinic conversion rate can only be estimated for the first three months of the deployment period due to otherwise missing data. The authors find conversion rates with magnitudes similar to estimates inferred for the East Greenland Current and the West Spitsbergen Current. Due to limitations in the mooring data set the authors have considered output from a high-resolution ROMS model. The respective analysis is part of an appendix to the paper. Therein, the authors aim at verifying how representative the mooring-derived energy conversion rates actually are. They conclude from the model analysis that the baroclinic energy conversion dominates over the barotropic energy conversion.

In general, the paper is written well enough. But I personally found it sometimes a bit tiring to read all the abbreviations. This is probably a matter of personal taste. I did wonder, however, why the model analysis was somewhat "hidden" in the appendix. The authors draw important conclusions from this model analysis. Any reader might easily miss the respective discussion by simply ignoring to read the appendix. Therefore, I think, this analysis deserves to be built into the main text.
The study of the authors contributes to improving the knowledge of one of the major currents transferring the warm and saline Atlantic Water towards the Arctic. I find that the manuscript addresses interesting scientific outcome on the nature of this current off the Lofoten Islands that is of interest to the readers of OS. The figures are generally of high quality. However, I partly missed information regarding the methods applied to the mooring time series. For example, it was several times mentioned that the mooring succumbed to "knock-down" events. But how these events were eliminated from the data remained unclear. There are other minor requests for clarification that I think will help to improve the manuscript further. Therefore, I recommend a minor revision of the manuscript.

My detailed comments are given below:

Page 1, line 25: the statement that "the front current is relatively poorly known" somehow contradicts the statements that follow in the next sentences. Therein, the authors quote several studies that provide transport estimates for the front current for various location. It might help to clarify what exactly is "relatively poorly known".

Page 3, line 36: please highlight the location of the Lofoten Escarpment in Figure 1 by adding a respective label. Same sentence starting with "there might...": there is a word missing, "be"?

Page 3, line 59: please add something like "based on the mooring records" at the end of the sentences

Page 4, Table 1: as there are different styles/cultures to write down dates, I suggest to write months using letters like May, Jun, Sep. This avoids that people mix up days and months.

Page 4, line 62: it remains unclear what kind of manuscript "Fer (2020)" actually is or where it can be assessed. The respective reference does not provide any relevant information. So, at present, any reader is not able to locate information on the data

set other than the one mentioned here. Same holds for page 5, lines 84/85, where the same reference is mentioned.

Page 4, line 80: previously, it was said that the used ADCPs were of type RDI 75 kHz Longranger. Now, they are addressed as RDI 75 kHz Sentinel Workhorse. To my knowledge, such a device operating at 75 kHz does not exist. According to the Teledyne-RDI web page, ADCPs of type Sentinel operate at frequencies >= 300 kHz and thus have a much shorter range than the Longranger ADCPs. Please, clarify.

Page 5, line 88: please, provide more information on the observed knock-down events, e.g. how often did they occur, how deep did the moorings descend, and how was this effect eliminated from the considered data ? Very much later (page 15), it is mentioned in the text that the data set was actually interpolated and gridded. This information and related specifics are missing here.

Page 5, line 100: please, refer here to the Copernicus Marine Environmental Monitoring Service (CMEMS) as the data provider, since there are still a number of papers out that still claim AVISO to be the data provider. Use of CMEMS data furthermore expects if not requires a proper credit of their data use, which is missing in this manuscript. My guess is that ECMWF expects something similar. Finally, as EKE is not a property provided as part of the used data product, how is EKE defined here ? Did you just consider the provided geostrophic anomalies ? The data set provides both, anomalies and absolute velocities. The present text is not clear enough on what kind of velocity fluctuations actually been used.

Page 5, line 116: please, add "2017" after "March"

page 7, line 135: it looks like the cross-component was also quite high in spring. In the upper part of the water column (Figure 4b) other seasons seem to be higher than winter or are of comparable magnitude. Could you comment on that?

Page 10, line 171: you could either repeat the separation distance of 6 km here, or otherwise provide readers with the size of the Rossby deformation radius at this location

page 10, lines 176/177: from figures 4 and 5 it is obvious that there isn't any velocity data at depth < ~200m at mooring MW. At times, there are data missing as deep as 300 m. Also MN does show data gaps for z < 200m. So, please, clarify how it is possible to infer the transport for the 50-650m range.

Page 10, line 179: how did you treat the temperature information outside the mooring array or during those times, when there wasn't any temperature information for mooring MW? Did you consider the depth of the 5°C isotherm to be constant across the entire width of the area used for calculating transports? Please, clarify as well.

page 10, line 190: please, clarify; relative to what reference level? Same line: "currents peak" or "the current peaks"?

Page 11, line 200: as the summer season is covered twice, is "summer" meant here as the average of both summer seasons ? Was there any difference between the two summers? One might guess so by looking at Figure 6.

page 12, Table 2: Please write Q_N and Q_S in the same way as it is used in the text and in the table, i.e. with small letters for "N" and "S". As will also be my question regarding Figure 7: as "annual" refers to the entire time series, and as this comprises two summer seasons, does this enter the uncertainties? Or asked differently, what is included in the uncertainties mentioned here?

Page 13, equation 1: if EKE is inferred from along-stream and across-stream velocities, it makes sense to keep the previously inferred terms u_a and u_x. Here and later in the text, the authors switch to u and v.

Page 13, line 241: please, provide a reference.

Page 14, lines 243/244: these lines need fixing. Furthermore, equation (2) does not contain a rho_0, which is part of equation (3), but a rho', which is not introduced. What reference density was used ? Shouldn't the right term of equation (2) be negative?

Page 15, lines 256: four times use of the word 'obtain'

page 15, lines 260, sentence starting with "The conversion rates calculated from. . .": please remove this entire sentence and the following as the information is identical to the one given in lines 279ff. There, it fits much better.

Page 15, line 269: the statement that the estimates from the Fram Strait are comparable to the Lofoten Escarpment is a bit tricky. The former values are O(100) mˆ2/sˆ2, the latter values are O(10ˆ-4) W/mˆ3. So, please, make the comparability more obvious to the reader.

Page 15, line 270: please introduce WSC

page 16, line 298: there is a word missing at the end of the line.

Figures:

Figure 1. Labels like "Norway" in Figure 1a and "NO" in Figure 1b are really hard to see. Think about adding a text label highlighting the location of the Lofoten Escarpment. The unit in the EKE colorbar should read 10ˆ-4 mˆ2sˆ-2, not 10ˆ4 mˆ2sˆ-2.

Figure A1. Both subplots lack a frame, at least in my printed version. Also the grid is almost invisible in the printed version. Maybe the authors can improve that. To the southwest of the red box, there is something like an arc-like pattern of very small-scale features in Fig A1a that look totally different from the remaining parts of the plot. What causes this?

---

## Author Comment (AC1) · 3 Apr 2020

**Referee #1, Anonymous**

This is our preliminary response to the reviewer's comments to encourage discussion when the discussion period is still open. We will provide a more complete response in our final response shortly after the discussion closes. The reviewer's comment is reproduced in black Calibri font followed by our response starting with Re in red, bold Arial font.

This work is based on a comprehensive data set, addresses important questions, and is generally well written and illustrated. Thus, I am confident that it deserves publication in something close to its present form, but it does contain some confusing aspects and details that need to be clarified before final acceptance, as elaborated below.

Re: We thank the reviewer for the detailed reading and constructive comments. We addressed all comments as detailed below.

**Main comments**

1) Velocity components: Figure 1b shows an (x,y) coordinate system and you expect velocity components to follow the standard notation (u,v). This is confirmed on lines 97-98: "Current components are along-slope, u, and across-slope, v", but then this statement is followed by: "(In the figures, we explicitly use the notation ua and ux, respectively.)" without explaining, which is which (and ux is not along the x-axis). Most of the manuscript seems to keep the (ua, ux) notation, but Sect. 7 with Eqs. (2) and (3) partly returns to the (u,v) notation. This is unnecessarily confusing. Stick to one notation. I would suggest (u,v).

Re: We agree that this was confusing. We realize we forgot to mention the notation,  $u_a$  for the along slope and  $u_x$  for the cross-slope component. We simply switch to the notation suggested by the reviewer because (x,y) defined on the map can be related to (u,v).

2) Projected distance: On lines 181-182, you write: "The moorings are separated by approximately 6 km (horizontal distance between the locations), and when projected onto the cross-slope section to their respective isobaths, the distance is about 8 km". How can a projected distance become larger than the distance ? Using the positions in Table 1, I get a distance of 5.3 km, which projected onto the cross-slope direction ought to be around 4 km. This projected distance is used extensively in the manuscript. I am not sure that any main results are substantially affected by this, but again it is unnecessarily confusing and has to be corrected.

Re: Thanks for pointing this out. First, agreed that the actual distance is 5.3 km (round up to 6 km is an error). The use of "projection" in this context was wrong. What we did was the following. We defined a cross-isobath section, normal to the isobaths (oriented 42° from East), through the position of MW covering between 100 m and 2500 m depth. We extracted the bathymetry for this section from ETOPO1. We calculated the cumulative distance between pairs of position on this section, giving us depth versus distance from the start of section. We obtained the distance of the moorings on the section by interpolation to the mooring isobaths of 1500 m and 655 m. The distance between them is then 8 km. The calculation is correct; however, inevitably includes uncertainties from the bathymetric data set in the steep escarpment. If projected to the cross-slope direction, because the relative angle between the mooring line orientation and the cross-isobath direction is about 20°, the projected distance is about 5 km (4 km the reviewer infers is for 42° relative orientation, which would be in error). We recalculated the transport estimates using a 5 km distance.

3) The width of the current for transport calculation: As I read your description (lines 182-187), your transport calculation is equivalent to multiplying the average transport density from the two moorings by 28 km (which according to the previous comment ought to be 24 km), but you do not mention what you do with the shallowing bottom in the 10 km inside of MN (to 250 m according to line 184). The statement on lines 184-185: "hence assign a 14 km effective width of water column to each mooring" does not indicate that this was taken into account. This should either be briefly clarified or the transport corrected if it has not been taken into account. As for many similar studies, the width of the current is probably the most uncertain aspect of the transport calculations, so is this uncertainty included in the uncertainties cited in Table 2 or are they just statistical (standard error ?). This should be clarified in the table heading and perhaps also in the text.

Re: Your interpretation is correct. We re-calculated accounting for the reduction is water crosssection area on the MN part. This effectively reduces the width to 7.6 km. The changes are as follows:

distance between two moorings = 5 (not 8) km

width of outer mooring is 12.5 km (2.5+10)

width of inner mooring is 7.6 km (2.5 + 5.1), calculated as an effective width to give the same surface area as the area when integrated using the actual topography to 10 km onshore of the 650 m isobath.

Table 2 of the original version listed the average and 1 standard deviation (we forgot to mention this in the caption, unfortunately). Now our error estimates are improved to also include standard error and all results are reported in the Table. Following up on reviewer 3's comment, we now also analyse summer 2016 and summer 2017 separately.

To calculate the standard error we use degrees of freedom (DOF) taking into account the decorrelation time scale of 7 days, already reported in the manuscript. DOF then equals number of daily data points in a window of interest divided by 7. The standard error, se, is calculated as the standard deviation, std, divided by (DOF)1/2. For example, for AW, the annual values are (mean, std, se): 2.0, 1.3, and 0.2 Sv.

In addition, we now calculate a representative transport error estimate, for winter, summer, and annual data points, separately. The procedure is complex, in order to account for the time variability in statistics. We assume about 20% error (4 km) in the effective width estimate, and assume 0.05 m/s error in depth averaged current at mooring (corresponding to 30 m2/s transport density). A simple calculation using these figures, ignoring the statistics, would lead to an error of 0.12 Sv. Using the mean and the std of observed transport density (for winter, summer and all data separately), we generate 100 random data points from a normal distribution (Uobs), and calculate the transport using 20.1 km width (Wobs) as Qobs = UobsxWobs. The distribution of transport is approximately normal in each season, and this assumed distribution for error analysis is justified. We then generate 100-point zero mean and imposed rms error data for transport density (Uerr) and width (Werr) from random distribution. Total transport (with error) is Qtot = (Wobs+Werr)\*(Uobs+Uerr). This is now 100 point time series of transport with error contribution. We calculate the residual, res = Qtot - Qobs, and bootstrap the rms of the residual 1000 times. The mean of the 1000 bootstrap error estimates is the transport error.

This results in an error of 0.8 Sv for annual averages, 0.7 Sv for summer and 0.9 Sv for winter. This is typically less than the standard deviation and 3-4 times the standard error.

4) Transport of the top 50 m layer: The average volume transport cited in the abstract (2.8 Sv) seems not to include the top 50 m (line 213). I assume the reason to be that this layer is less saline than 35.17 on average (Figure 2) due to some admixture of water from the Norwegian Coastal Current, but isn't the fraction of Atlantic water in this layer still » 50% ? Using S=35.17 as a lower boundary for Atlantic water does not necessarily imply that you should use the same criterion for an upper boundary. If you want to retain this, it should in any case be better justified in the text.

Re: This is a very good point. The reason we excluded the top layer was two fold: the low salinity layer as the reviewer noted but also, more importantly, the lack of measurements. Note that we already report (in our sensitivity analysis) an estimate of the increase in transport when the uppermost measurements are extended to the surface, and all water is assumed AW in the top 50 m. See line 213 of the original manuscript "Including the top 50 m increases the total mean transport by 0.4 Sv (from 2.8 Sv)." With revised calculations, it increases by 0.3 Sv (from 2.0 Sv).

We have examined the upper layer from a freely accessible hydrological Altas of the Nordic Seas (Bosse and Fer (2018) Hydrography of the Nordic Seas, 2000-2017: A merged product https://doi.org/10.21335/NMDC-1131411242). The top 50 m layer at MW, corresponding to the core of the slope current, varies from about 35.25 to 34.95 g/kg (figure R1). Assuming values of shelf water of

Figure R1: Monthly mean profiles of absolute salinity at each mooring position considering profiles from the hydrographical atlas at less than 25km and 250m bathymetric difference. Black contours show selected isopycnals. Minimum values for MN, MS, MW and MB are respectively 34.21, 34.75,34.95, 35.22 g/kg.

**Details**

5) line 7: "volume transport" -> "volume transport of Atlantic water"

**Re: corrected**

6) line 19: "Iceland-Faroe . . . channels" is not standard. You could use "gaps" instead of "channels"

**Re: corrected**

7) line 110: "Norwegian Sea Deep Water" -> "deeper water". The NSDW is usually reserved for the deepest component, which is separated from AW by intermediate water masses.

**Re: corrected**

8) Figure 2, third line of caption: "typical value" -> "typical lower limit".

**Re: corrected**

9) lines 118-119 : The last sentence seems to contradict the previous sentence. line 168: "the two moorings" -> "MS and MN".

**Re: corrected**

10) lines 175 onwards : In line 175, the letter "Q" (with or without subscripts) is defined as "transport density (m\*\*2/s). After that, the same letter is used for "transport" (m\*\*3/s). This should be corrected.

**Re: corrected**

11) Figure 8: Consistent with my main comment 1) above, I suggest that you use (Wx,Wy) instead of (Wa,Wx). Having Wx perpendicular to the x-axis is confusing. In the caption to this figure, it might also be emphasized that the velocities in Figure 8b (presumably) are low-passed (not band-passed, as might be assumed from line 220).

**Re: corrected (using Wx and Wy).**

12) line 324: Isn't there a rho-0 too much in the last equation ?

Re: corrected. This was a typo and the conversion rate calculations are correct.

13) line 330: Isn't a parenthesis missing in the middle equation ?

**Re: corrected**

14) Figure A1b: In my printed version of the manuscript, I could not identify any "magenta line". On the screen, there is something that could be magenta, but it does not resemble a line.

Re: We improved the presentation of Fig A1.

15) Appendix A: The contents of the appendix seem rather crucial to some of the results and the last parts of the abstract. You might consider pulling it into the main manuscript as a separate section just before the conclusion.

Re: We now incorporated the modelling part into the body of the manuscript.

---

## Author Comment (AC2) · 3 Apr 2020

**Referee #2, Michael A. Spall**

**This is our preliminary response to the reviewer's comments to encourage discussion when the discussion period is still open. We will provide a more complete response in our final response shortly after the discussion closes. The reviewer's comment is reproduced** in black Calibri font **followed by our response starting with Re. in red, bold Arial font.**

This is a timely analysis of mooring data within the Norwegian Atlantic Slope Current on the eastern side of the Lofoten Basin. This region has been identified as a source of eddy kinetic energy and offshore eddy heat flux, which is important for the basinscale stratification and air-sea exchange. The analysis is fairly straightforward and I recommend that it be published subject to relatively minor revisions. There are a couple suggestions for additional analysis that, while not crucial, would provide more context for the results.

**Re. We thank the reviewer for the detailed reading and constructive comments. We addressed all comments as detailed below. In response to the reviewer's comments, we aim to include two new figures (Figs. R1 and R2) in the revised version as described below.**

Would it be possible to compare the transport in density and depth with that at the Svinoy section? If the transport there is barotropic, and here it is baroclinic, that implies that there has been some upwelling between these two stations, or a loss of transport in the deeper layers. If the transports are similar, can you tell if the isopycnals have risen or if there has been a water mass transformation between these two sections? I think a more complete comparison with that upstream section can reveal more about what has happened between these locations. Even if the years are different, maybe you can consider the seasonal cycle, which should be representative.

**Re. There is indeed substantial AW transformation between the Svinøy and Gimsøy sections. This was the subject of a previous paper (Bosse et al, JGR 2018). Analyses on a T-S diagram and in isopycnal layers showed that AW was progressively transformed to denser isopycnals. This generated a poleward warming in density surfaces just below the AW at Svinøy (a signal already shown by Rossby et al, DSR 2009). While the most important transformation occurred in the western part of the Lofoten Basin, lateral exchanges generated by instabilities of the slope current substantially modified the characteristics of the AW transported from the Svinøy to Gimsøy section.**

**In order to examine the vertical structure of the slope current, we have constructed mean T-S sections along the Svinøy and Gimsøy sections using freely accessible dataset of the Nordic Seas (Bosse and Fer (2018) Hydrography of the Nordic Seas, 2000-2017: A merged product https://doi.org/10.21335/NMDC-1131411242), see figure R1. We will include this figure (Fig R1) in the revised manuscript.**

**About isopycnal displacements: isopycnals with $\sigma_0 < 27.7$ kg m$^{-3}$ rise due to AW transformation peaking at the 27.5 isopycnal with a displacement of 150 m at the core of the slope current. This isopycnal switches from being located below the AW core to above. This is also where the largest spiciness injection by vertical mixing was reported by Bosse et al (2018). In deeper layers, we observe a sinking of isopycnal from Svinøy to Gimsøy, which can be explained by the presence of other intermediate waters subducted along the Mohn Ridge Front and AW transformations in the Lofoten Basin, decreasing the stratification in the AW pycnocline.**

[Figure]

**Figure 1:** Mean (a-c) temperature and (b-d) salinity cross-front sections taken along the Gimsøy and Svinøy sections. Isopycnals are shown by black contours. A map shows all profiles used, located within 25 km distance from the sections. The annual mean was constructed by averaging over four seasonal sections. Each section is obtained by binning profiles projected onto the section in 5 km cross-section intervals. The seasonal sections for temperature and salinity were smoothed using a Gaussian moving window of 10 km variance prior to annual averaging.

Can the authors provide error bars for the velocity and transport estimates?

**Re. We now provide error bars for the velocity and transport estimates: for winter profiles in Fig 4 (for both the temperature and velocity profiles), for monthly averaged AW transport in Fig 6b, for the annual average in T-binned histogram of Fig. 7, and in Table 2 where we list the average transports. We will post the updated material in the revised manuscript after the discussion.**

**In the figures, the error bars are based on the standard error, se = std / sqrt(DOF), using the standard deviation (std) over the analysis period and using the effective degrees of freedom, DOF, estimated as the number of observation points divided by the decorrelation length scale of 7 days (this was estimated in the manuscript). For example, for the winter averages using daily profiles, the standard deviation is over the number of winter days (90), and DOF = 13.**

**In addition to the standard error, we estimate a representative error for the transport (including error estimate from the width and depth-averaged current). Table 2 will be updated with errors calculated for each analysis period (annual, summer and winter; and summer 2016 and summer 2017, separately). More details in error calculations can be found in our response to reviewer 1.**

Introduction:

You might also reference Clark and Straneo (Observations of Water, Mass Transformation and Eddies in the Lofoten Basin of the Nordic Seas, JPO, 2015).

**Re: Done. We normally cite Clark Richards & Straneo in our Lofoten Basin papers. Unfortunately it was overlooked in this manuscript.**

I had a 2010 paper in Ocean Modeling that would be more appropriate to reference than the 2010 JPO paper as it addresses the lateral eddy heat flux in (an idealized) Lofoten Basin (Spall,Non-local topographic influences on deep convection: An idealized model for the Nordic Seas, Ocean Modeling, 32, 72-85).

**Re. Thank you for point this reference out. Changed as suggested.**

lines 120-124: It should be possible to quantify the source of the increased vertical shear, or at least break it down into temperature and haline contributions via thermal wind.

**Re. We calculated the vertical shear of geostrophic current perpendicular to Gimsøy and Svinøy sections (Figure R2), derived from the Nordic Seas data set described above. To compare the baroclinicity of the slope current at these two locations, we vertically integrated the shear with a level of no motion at the bottom. This allows to confirm the increase in baroclinicity of the slope current with latitude (63 cm/s at Gimsøy vs 46 cm/s at Svinøy, once vertically integrated).**

[Figure]

**Figure R2: Vertical shear from thermal wind balance for (a-b-c) Gimsøy and (d-e-f) Svinøy sections. Panels a&d show the total shear, and b&e the thermal and c&f the haline contribution. In each plot grey contours are isopycnals in a&d, isotherms in b&e and isohalines in c&f. The black contours correspond to the bottom to surface integrated vertical shear in m/s.**

In particular, it is interesting to note that the contribution of temperature to shear is actually stronger south at Svinøy (75 cm/s vs 56 cm/s once integrated), but counter-balanced by a strong negative contribution of salinity reaching -31 cm/s integrated from bottom to 150m and -25cm/s to the surface (due to the presence of a deep patch of more saline AW at the slope). The negative shear due to salinity only reaches -12 cm/s from the bottom to 250 m and become insignificant when integrated to the surface.

There is a weaker signature of AW in T-S at Gimsøy, so the corresponding positive and negative shear contributions to geostrophic currents in the slope current are weaker there. Furthermore, a stronger interaction of the coastal current with the slope current at Gimsøy (because of proximity of the slope current to the coast due to steep slope) contributes positively (i.e., poleward) to the transport than at Svinøy. Note that the broader region of isopycnal gradients at the Svinøy slope does not necessarily imply a broader current than for Gimsøy, but could be due to a more variable position linked to the steepness of the slope constraining the circulation.

We will include this figure and the related discussion in the revised manuscript.

line 157: It might be useful to provide a scaling for the expected response to changes in the wind stress. One could calculate the onshore Ekman transport, downward deflection of the isopycnals, and the geostrophic response. The paper by Choboter et al. (2011, Exact Solutions of Wind-Driven Coastal Upwelling and Downwelling over Sloping Topography, JPO, 41, 1277-1295) provides analytic solutions but you might be able to do something useful just with simple scaling.

Re. This is not addressed yet at the time of this response.

line 191: It seems likely that the transport variability is due to the current meandering outside the moorings (rather than a change in the along-slope transport), but this isn't explicitly mentioned..

Re: Agreed. We now mention this point.

Figure 7: I found this to be the most surprising part of the paper. Any ideas why there is more warm water in winter than in summer? When/where was this water last exposed to the atmosphere? Was this subducted in the previous summer? If you see the same phase at Svinoy, which is O(1000 km) upstream, that would argue against it simply being advected along the slope. I think some more discussion around this finding would be helpful. The penetration of AW down to 650 m depth is likely related to that being the sill depth upstream.

Re. The increase of the temperature might of different origins, including the increase of AW thickness observed in winter at the mooring position as vertical mixing occurs; a seasonal peak in temperature; or the absence of moored observations in the upper 80 m where most of the seasonal warming is observed.

In Fig R3 we plot the time series of depth-averaged along-isobath current and temperature at Svinøy and the Lofoten moorings, MN and MW. The records are from the same period starting from June 2016. The Svinøy data are from the S1 mooring, kindly provided by Kjell Arild Orvik, and include hourly time series from RCM7 current meters at 100, 300 and 490 m depth. A seasonal signal in temperature is clearly observed at Svinøy. The pattern is similar at Gimsøy, and the winter temperature anomaly is larger. There is no apparent phase difference. The temperature anomaly in January observed at MW is not seen at Svinøy and cannot be advection. Note the largest current in January (same time as this temperature peak), detected in all moorings with no phase lag.

[Figure]

**Figure R3. Vertically averaged along-isobath current and temperature anomaly at Svinøy mooring S1 (Sv), and Lofoten moorings MN and MW. S1 data are kindly provided by Kjell Arild Orvik and used with permission. S1 data include hourly time series from RCM7 current meters at 100, 300 and 490 m depth (20 m above bottom). Vertical averaging is 100 m to 20 m above bottom for Sv and MN, and to 600 m depth for MW. The indicated time-averaged values are removed from the temperature records (8C from Sv and 6.5C from MN and MW).**

**Note that Fig R3 vertical averaging starts from 100 m depth. Vertical averages from the hydrography, averaged between surface and 500 m depth using monthly climatology at the mooring locations are shown in Fig R4. This somewhat supports the annual cycle with warm water in winter. Unfortunately, Gimsøy time series lack winter data. Svinøy temperatures are larger in fall and winter than in summer consistent with Fig R3b.**

**Overall, we interpret the largest warm water transport in winter as a consequence of the annual cycle of depth-averaged temperature coinciding with the time of strongest currents (winter).**

**Figures R3 and R4 will not be used in the revised manuscript. We will probably not discuss these points**

[Figure]

**Figure R4. Temperature and salinity anomalies (annual mean removed) at different sites. Vertical averaging is from 0 to 500m. Svinøy600 and Gimsøy600 can be directly compared to Sv and MN. W is at the position of MW (and B is at the position of MB in the Lofoten basin).**

line 232: BC and BT were also calculated from a high resolution mooring array in Spall et al. (2008).

**Re: We now include this in our list of examples.**

line 238-239: CHECK!NOT 1 MONTH?

**Re: Apologies for this confusing "note-to-self", which we forgot to remove (we had posted a short comment on this later).**

lines 242-245: This justification is not very convincing, I suggest deleting it.

**Re: Deleted as suggested.**

line 340: Magenta does not stand out compared to the colorbar, I suggest using a different color to mark the line.

**Re: We improved this figure.**

---

## Author Comment (AC3) · 3 Apr 2020

**Referee #3, Anonymous**

**This is our preliminary response to the reviewer's comments to encourage discussion when the discussion period is still open. We will provide a more complete response in our final response shortly after the discussion closes. The reviewer's comment is reproduced** in black Calibri font **followed by our response starting with Re in red, bold Arial font.**

Overview and general recommendation:

The manuscript describes the outcome of a mooring effort, carried out between Jun 2016 and August 2017 across the Norwegian Atlantic Slope Current off the Lofoten Islands at the so-called Lofoten Escarpment. The authors exploit the data from a mooring array that consisted of three deep sea moorings. Two of them, moorings MN and MW, were located about 6 km apart from each other across the slope current. A third mooring, MS was located almost 30 km further upstream close to the Gimsøy hydrographic repeat section. A fourth mooring, MB, was located in the interior of the Lofoten Basin and was not part of the analysis. The authors use the mooring records, mainly velocity data obtained from Longranger ADCPs as well as T/S information from MicroCATs or temperature loggers, to address the Atlantic Water (AW) layer within the Norwegian Atlantic Slope Current that is captured by the moorings. While there are already descriptions of the Slope Current from the sections located upstream, the authors state that they provide the first mooring based description for the Gimsøy region off the Lofoten Islands. The authors describe the general nature of the velocity structure in the upper water column and find the strongest velocities in the winter period. This timing coincides with the time of the warmest temperatures observed in the AW layer. The authors furthermore infer transport time series for the two moorings MN and MW, explain their choice of a respective area over which the transport is calculated and finally quantify the volume transport for the AW layer. The authors furthermore address the forcing and find a correspondence between the along-stream wind forcing and the along-stream current component. Finally, the authors infer energy conversion rates from the mooring records, in particular baroclinic and barotropic conversion rates that describe the transfer of mean potential energy into eddy kinetic energy and the transfer of mean kinetic to eddy kinetic energy. The baroclinic conversion rate can only be estimated for the first three months of the deployment period due to otherwise missing data. The authors find conversion rates with magnitudes similar to estimates inferred for the East Greenland Current and the West Spitsbergen Current. Due to limitations in the mooring data set the authors have considered output from a high-resolution ROMS model. The respective analysis is part of an appendix to the paper. Therein, the authors aim at verifying how representative the mooring-derived energy conversion rates actually are. They conclude from the model analysis that the baroclinic energy conversion dominates over the barotropic energy conversion.

**Re: We thank the reviewer for the detailed reading and constructive comments. We addressed all comments as detailed below. We do not respond to the above text, which is a nice summary of our paper (note we edited Grinvoy to Gimsøy in the above copy).**

In general, the paper is written well enough. But I personally found it sometimes a bit tiring to read all the abbreviations. This is probably a matter of personal taste. I did wonder, however, why the model analysis was somewhat "hidden" in the appendix. The authors draw important conclusions from this model analysis. Any reader might easily miss the respective discussion by simply ignoring to read the appendix. Therefore, I think, this analysis deserves to be built into the main text.

**Re. Thank you for this suggestion. We agree. We integrated the Appendix to the main body of the revised manuscript. We also attempt to remove some abbreviations.**

The study of the authors contributes to improving the knowledge of one of the major currents transferring the warm and saline Atlantic Water towards the Arctic. I find that the manuscript addresses interesting scientific outcome on the nature of this current off the Lofoten Islands that is of interest to the readers of OS. The figures are generally of high quality. However, I partly missed information regarding the methods applied to the mooring time series. For example, it was several times mentioned that the mooring succumbed to "knock-down" events. But how these events were eliminated from the data remained unclear. There are other minor requests for clarification that I think will help to improve the manuscript further. Therefore, I recommend a minor revision of the manuscript.

**Re. Thank you for your assessment of our manuscript. We made minor revisions and clarifications to address your detailed comments below. We also improved the description of the methods applied to the mooring data. Fer (2020) (see also response below) is the mooring data set together with a detailed data report. The data set and the report are openly accessible with CC BY 4.0 license. Unfortunately, neither the URL, DOI or the fact that this is a data set was apparent in the citation. We did not notice this. This reference (data and the data report) was a reason why we kept the description of data treatment relatively concise. We now corrected the citation and also include the URL in the data statement.**

My detailed comments are given below:

Page 1, line 25: the statement that "the front current is relatively poorly known" somehow contradicts the statements that follow in the next sentences. Therein, the authors quote several studies that provide transport estimates for the front current for various location. It might help to clarify what exactly is "relatively poorly known".

**Re. We replaced this opening sentence with "The front current, which is not addressed in this study, has not been measured in detail using current meter arrays, but geostrophic transport estimates are available from hydrography."**

Page 3, line 36: please highlight the location of the Lofoten Escarpment in Figure 1 by adding a respective label. Same sentence starting with "there might. . .": there is a word missing, "be"?

**Re. Done**

Page 3, line 59: please add something like "based on the mooring records" at the end of the sentences

**Re. Done**

Page 4, Table 1: as there are different styles/cultures to write down dates, I suggest to write months using letters like May, Jun, Sep. This avoids that people mix up days and months.

**Re. Done**

Page 4, line 62: it remains unclear what kind of manuscript "Fer (2020)" actually is or where it can be assessed. The respective reference does not provide any relevant information. So, at present, any reader is not able to locate information on the data set other than the one mentioned here. Same holds for page 5, lines 84/85, where the same reference is mentioned.

**Re. Thanks for noticing this. Unfortunately, Fer (2020) was not formatted properly. It is the mooring data set together with a detailed data report, openly accessible with CC BY 4.0 license from the Norwegian Marine Data Centre. Neither the URL, DOI or the fact that this is a data set, was apparent in the citation (because of the formatting). We did not notice this. We now corrected and also updated the data availability statement with the link.**

Page 4, line 80: previously, it was said that the used ADCPs were of type RDI 75 kHz Longranger. Now, they are addressed as RDI 75 kHz Sentinel Workhorse. To my knowledge, such a device operating at 75 kHz does not exist. According to the Teledyne-RDI web page, ADCPs of type Sentinel operate at frequencies >= 300 kHz and thus have a much shorter range than the Longranger ADCPs. Please, clarify.

**Re. The reviewer is correct. "75 kHz Sentinel" is a mistake and now corrected. In MN, MS and MW we used only Longranger ADCPs. (Additional 300 kHz Sentinels were also deployed in MB, but not reported in this paper.)**

Page 5, line 88: please, provide more information on the observed knock-down events, e.g. how often did they occur, how deep did the moorings descend, and how was this effect eliminated from the considered data ? Very much later (page 15), it is mentioned in the text that the data set was actually interpolated and gridded. This information and related specifics are missing here.

**Re. We provide some description now, but refer the reader to the data report for a detailed description. Specifically, pressure time series are shown in the report, and inform about the knock down events.**

**About interpolation and gridding: Data from all instruments are first averaged into one hour intervals (if the sampling rate was faster) and then interpolated to a common 1-hour time stamp. The time variable depth (pressure) records were constructed at each time stamp and for each instrument using vertical interpolation of the known target depth (of instruments with pressure sensor) and the measured pressure to the target depths of all instruments. Hourly profiles of temperature, salinity and horizontal current were then vertically interpolated to 10-m vertical resolution. A depth level with a data coverage less than 30% of the total measurement duration was excluded. This 1-hour-10 m vertical homogeneous, gridded data matrix was cleaned from short segments of data (especially in the outer ranges of the ADCPs) by filling with NaNs when a duration of segment with data was less than 3 days.**

Page 5, line 100: please, refer here to the Copernicus Marine Environmental Monitoring Service (CMEMS) as the data provider, since there are still a number of papers out that still claim AVISO to be the data provider. Use of CMEMS data furthermore expects if not requires a proper credit of their data use, which is missing in this manuscript. My guess is that ECMWF expects something similar. Finally, as EKE is not a property provided as part of the used data product, how is EKE defined here ? Did you just consider the provided geostrophic anomalies ? The data set provides both, anomalies and absolute velocities. The present text is not clear enough on what kind of velocity fluctuations actually been used.

**Re. We now refer to E.U. Copernicus Climate Change Service and E.U. Copernicus Marine Service Information as the data provider, and properly cite ERA-5 and ERA-Interim. We calculated the EKE using the geostrophic current anomalies obtained from the sea-level anomaly. This is now clarified in the revised version.**

Page 5, line 116: please, add "2017" after "March".

**Re. Done**

page 7, line 135: it looks like the cross-component was also quite high in spring. In the upper part of the water column (Figure 4b) other seasons seem to be higher than winter or are of comparable magnitude. Could you comment on that?

**Re. We revised this part as: "Cross-slope component was weak (typically ± 0.02 m s$^{-1}$) and increased in spring and winter, with largest 200-600m depth-averaged values in winter (0.05 m s$^{-1}$ at MW) and an increased variability with depth (Figure 4b, d). In the upper part of the water column at MW, averaged cross-slope velocities in fall exceeded the winter values. This is consistent with**

**the increased $EKE_g$ at MW location, calculated from satellite measurements in November 2016 (Figure 3c)."**

Page 10, line 171: you could either repeat the separation distance of 6 km here, or otherwise provide readers with the size of the Rossby deformation radius at this location

**Re. We repeat the separation distance (note that this is now reduced to 5 km; see response to reviewer 1).**

page 10, lines 176/177: from figures 4 and 5 it is obvious that there isn't any velocity data at depth < 200m at mooring MW. At times, there are data missing as deep as 300 m. Also MN does show data gaps for z < 200m. So, please, clarify how it is possible to infer the transport for the 50-650m range.

**Re. The shallowest available measurement is extended upward to 50 m. This is now clarified.**

**At MW, velocity measurements are limited in the vertical. The data gap is 18% at 250 m depth and rapidly increases to 60% and 85% at 200 and 190 m. Temperature at MW is better covered with 20% gap at 90 m, increasing to 70% at 80 m. Velocity measurements at MN have 35% gap at 150 m, increasing to 50% at 80 m.**

Page 10, line 179: how did you treat the temperature information outside the mooring array or during those times, when there wasn't any temperature information for mooring MW? Did you consider the depth of the 5C isotherm to be constant across the entire width of the area used for calculating transports? Please, clarify as well.

**Re. For full-duration transport calculations, we used the temperature record from MW, which is available for the entire deployment and is well-resolved in the vertical. We extended the uppermost temperature measurement to 50 m when there were gaps in the data (typically above 90 m depth, see above). These points are now clarified.**

page 10, line 190: please, clarify; relative to what reference level? Same line: "currents peak" or "the current peaks"?

**Re. clarified and corrected: relative to surface pressure; curent peaks**

Page 11, line 200: as the summer season is covered twice, is "summer" meant here as the average of both summer seasons? Was there any difference between the two summers? One might guess so by looking at Figure 6.

**Re. Yes, we averaged both summers. We now also report and discuss the differences between summer 2016 and 2017. We do not show the mean profiles separately because Fig 4 is too crowded. In summary (using only depth levels where there's more than 70% data for each season):**

**Summer 2016 and summer 2017 averaged temperature profiles at MW are very similar, equal to within 0.5C in the upper 600 m and identical in deeper layers.**

**At MW, u is about 1 cm/s larger below 300 m (a barotropic increase) in summer 2017, and shear is stronger in the top 300m, increasing by 6 cm /s to 200 m depth.**

**At MN, below 400 m (bottom 250 m) u is the same, but shear is stronger in summer 2017 higher in the water column, with u increasing by an additional 10 cm/s to 200 m depth.**

**Transport is stronger in summer 2017 relative to 2016. Average values, standard deviation and standard errors are now listed separately for each summer in Table 2. (Using the revised width calculations in response to reviewer 1, average Q (± standard error) increases from 1.4 (±0.2) Sv in summer 2016 to 1.9 (±0.3) Sv in summer 2017).**

page 12, Table 2: Please write Q_N and Q_S in the same way as it is used in the text and in the table, i.e. with small letters for "N" and "S". As will also be my question regarding Figure 7: as "annual" refers to the entire time series, and as this comprises two summer seasons, does this enter the uncertainties? Or asked differently, what is included in the uncertainties mentioned here?

**Re. Corrected (we are now using p(ositive) and n(egative)). The averaging durations, standard deviations, degrees of freedom and standard errors are now calculated and clarified in the table and in the text.**

Page 13, equation 1: if EKE is inferred from along-stream and across-stream velocities, it makes sense to keep the previously inferred terms u_a and u_x. Here and later in the text, the authors switch to u and v.

**Re. We now use u and v (along and across isobath components) throughout.**

Page 13, line 241: please, provide a reference.

**Re. Inserted Spall et al. (2008).**

Page 14, lines 243/244: these lines need fixing. Furthermore, equation (2) does not contain a rho_0, which is part of equation (3), but a rho', which is not introduced. What reference density was used ? Shouldn't the right term of equation (2) be negative?

**Re. Corrected, and all these points now clarified. We used $\rho_0$ = 1027 kg m$^{-3}$. The sign of Eq(2) is correct in the convention we described (it can also be obtained by simplifying Eq. A1; note the sign of the last term.).**

Page 15, lines 256: four times use of the word 'obtain'

**Re. Improved the text.**

page 15, lines 260, sentence starting with "The conversion rates calculated from. . .": please remove this entire sentence and the following as the information is identical to the one given in lines 279ff. There, it fits much better.

**Re. Done.**

Page 15, line 269: the statement that the estimates from the Fram Strait are comparable to the Lofoten Escarpment is a bit tricky. The former values are O(100) m^2/s^2, the latter values are O(10^-4) W/m^3. So, please, make the comparability more obvious to the reader.

**Re. The comparison starts with the EKE density. The values are (50-200)×10$^{-4}$ m$^2$ s$^{-2}$ (both in Fram Strait and the Lofoten slope). Then we compare the conversion rates (BT and BC) which are O(10$^{-4}$) W m$^{-3}$ both in Fram Strait and the Lofoten slope. Perhaps the transition from EKE to BT (and BC) is too abrupt? We smoothened this by rewording. We also inserted (in line 265) the following to better interpret the two parameters.**

**"For reference, a conversion rate of 10$^{-4}$ W m$^{-3}$ for 1 day accounts for $\rho_0$EKE of O(10) J m$^{-3}$, or EKE of O(100)×10$^{-4}$ m$^2$ s$^{-2}$."**

Page 15, line 270: please introduce WSC

**Re. We decided to remove this abbreviation because it is used only a couple of times.**

page 16, line 298: there is a word missing at the end of the line.

**Re. Corrected.**

Figures:

Figure 1. Labels like "Norway" in Figure 1a and "NO" in Figure 1b are really hard to see. Think about adding a text label highlighting the location of the Lofoten Escarpment. The unit in the EKE colorbar should read 10ˆ-4 mˆ2sˆ-2, not 10ˆ4 mˆ2sˆ-2.

**Re. Done**

Figure A1. Both subplots lack a frame, at least in my printed version. Also the grid is almost invisible in the printed version. Maybe the authors can improve that. To the southwest of the red box, there is something like an arc-like pattern of very small-scale features in Fig A1a that look totally different from the remaining parts of the plot. What causes this?

**Re. Thanks for pointing it out. We improved Fig A1. Among other improvements (projection, zoom, representation of the transect), the figure now includes a frame and a visible grid.**

**Regarding the small-scale features mentioned, we observed that these only occur during fall and that they are related to noisy horizontal density gradients. The features are mainly confined between the 250 and 400 m isobaths, and particularly in the southern region of the domain, and only in fall. The isobath range and the region where this "noise" occur are not relevant for our conclusions; therefore we removed them without commenting in the text.**

---

## Referee Comment (RC4) · Anonymous Referee #1 · 5 Apr 2020

Dear Ilker Fer

Yes, this is one of the nice things about Ocean Science, that you can discuss topics in a less formal way than with the traditional review process.

But, I feel that you have addressed all my comments thoroughly and I have no further concerns.

Congratulations on a job well done

---

## Author Comment (AC4) · 26 Apr 2020

Thank you! No action taken. I am posting this as it is required by the system to respond to all comments before I can post the final author's response.
* * *

---

## Author Response (AR1)

**Final author response to the reviewers' comments on Norwegian Atlantic Slope Current along the Lofoten Escarpment" by Fer et al.**

In this final response, we merge the 3 responses we provided during the discussion stage, and make further adjustments and edits to reflect the changes made in the final revised version of our discussion paper. The reviewers' comments are reproduced in black Calibri font followed by our response starting with "Re:" in red, bold Arial font. We thank all reviewers for the detailed reading and constructive comments.

A brief summary of the major changes made is as follows. More details can be found in our point-by-point response below, and in the marked-up version of the revised manuscript. In some response below, we insert the exact change made by pasting snipped images from the manuscript.

Figures are improved:

      Fig. 1: edited for legibility, place names and some minor corrections

      Fig. 2: horizontal axis is now referenced to the 500 m isobath to be consistent with the Gimsøy and Svinøy climatology sections (new Figs 8 and 9)

      Fig. 4: Error bars are shown for the winter profiles

      Fig. 6 and 7 (and Table 2): Error bars are shown

      New Fig 8: Annual mean hydrography in Gimsøy and Svinøy sections using the Nordic Seas data set (Bosse and Fer, 2018)

      New Fig 9: Geostrophic shear and the thermal and haline contributions from the annual mean hydrography in Gimsøy and Svinøy sections using the Nordic Seas data set.

      Fig 10. Edited annotations and line colors for consistency

      Fig 11. Improved with new projection and edited for legibility. Removed noise from panel a.

      Fig 12. Edited presentation and line colors for consistency.

We integrated the conversion rate calculations from the ROMS output into the main body of the paper (previously it was in an appendix).

We have a new sub-section "2.2. Other Data", where we now describe the environmental forcing data sets, the Nordic Seas hydrography data set (Bosse and Fer, 2018) used in the revised version, and the ROMS fields. This makes the paper better organized.

We inserted error bars in several figures and the transport table. We amended a paragraph in "Section 6. Transport" where details of transport error calculation are described. We corrected our mooring effective width for transport calculations and revised the transport estimates.

We inserted a new section (7. Climatological structure and comparison with the Svinøy section) with two new figures (Fig 8 and 9).

**Referee #1, Anonymous**

This work is based on a comprehensive data set, addresses important questions, and is generally well written and illustrated. Thus, I am confident that it deserves publication in something close to its present form, but it does contain some confusing aspects and details that need to be clarified before final acceptance, as elaborated below.

Re: We thank the reviewer for the detailed reading and constructive comments. We addressed all comments as detailed below.

Main comments

1) Velocity components: Figure 1b shows an (x,y) coordinate system and you expect velocity components to follow the standard notation (u,v). This is confirmed on lines 97-98: "Current components are along-slope, u, and across-slope, v", but then this statement is followed by: "(In the figures, we explicitly use the notation ua and ux, respectively.)" without explaining, which is which (and ux is not along the x-axis). Most of the manuscript seems to keep the (ua, ux) notation, but Sect. 7 with Eqs. (2) and (3) partly returns to the (u,v) notation. This is unnecessarily confusing. Stick to one notation. I would suggest (u,v).

Re: We agree that this was confusing. We realize we forgot to mention the notation, $u_a$ for the along slope and $u_x$ for the cross-slope component. We switch to the notation suggested by the reviewer because (x,y) defined on the map can be related to (u,v).

2) Projected distance: On lines 181-182, you write: "The moorings are separated by approximately 6 km (horizontal distance between the locations), and when projected onto the cross-slope section to their respective isobaths, the distance is about 8 km". How can a projected distance become larger than the distance ? Using the positions in Table 1, I get a distance of 5.3 km, which projected onto the cross-slope direction ought to be around 4 km. This projected distance is used extensively in the manuscript. I am not sure that any main results are substantially affected by this, but again it is unnecessarily confusing and has to be corrected.

Re: Thanks for pointing this out. First, agreed that the actual distance is 5.3 km (round up to 6 km is an error). The use of "projection" in this context was wrong. What we did was the following. We defined a cross-isobath section, normal to the isobaths (oriented 42° from East), through the position of MW covering between 100 m and 2500 m depth. We extracted the bathymetry for this section from ETOPO1. We calculated the cumulative distance between pairs of position on this section, giving us depth versus distance from the start of section. We obtained the distance of the moorings on the section by interpolation to the mooring isobaths of 1500 m and 655 m. The distance between them is then 8 km. The calculation is correct; however, inevitably includes uncertainties from the bathymetric data set in the steep escarpment. If projected to the cross-slope direction, because the relative angle between the mooring line orientation and the cross-isobath direction is about 20°, the projected distance is about 5 km (4 km the reviewer infers is for 42° relative orientation, which would be in error). We recalculated the transport estimates using a 5 km distance.

3) The width of the current for transport calculation: As I read your description (lines 182-187), your transport calculation is equivalent to multiplying the average transport density from the two moorings by 28 km (which according to the previous comment ought to be 24 km), but you do not mention what you do with the shallowing bottom in the 10 km inside of MN (to 250 m according to line 184). The statement on lines 184-185: "hence assign a 14 km effective width of water column to each mooring"

does not indicate that this was taken into account. This should either be briefly clarified or the transport corrected if it has not been taken into account. As for many similar studies, the width of the current is probably the most uncertain aspect of the transport calculations, so is this uncertainty included in the uncertainties cited in Table 2 or are they just statistical (standard error ?). This should be clarified in the table heading and perhaps also in the text.

**Re: Your interpretation is correct. We re-calculated accounting for the reduction in water cross-section area on the MN part. This effectively reduces the width to 7.6 km. The changes are as follows:**

> **distance between two moorings = 5 (not 8) km**
> **width of outer mooring is 12.5 km (2.5+10)**
> **width of inner mooring is 7.6 km (2.5 + 5.1), calculated as an effective width to give the same surface area as the area when integrated using the actual topography to 10 km onshore of the 650 m isobath.**

**We inserted the following in section 6:**

The moorings MN and MW are separated by 5.3 km (horizontal distance between the locations), and when projected onto the cross-slope section the distance is about 5 km (the relative angle between the mooring line orientation and the cross-isobath direction is 20°). We assume velocity measured at each mooring is representative for the half-width (2.5 km) to the next mooring. We further extend the width of MW 10 km off-slope (distance to the 2500 m isobath) and MN 10 km on-shore (distance to the 250 m isobath). These choices are motivated by the coverage of the dynamic AW core at Gimsøy section (see Fig. 2). The outer edge corresponds to the location where the 5°C isotherm is shallowest, and covers the relatively steep lateral isopycnal gradient toward the slope. The width of water column for the outer mooring MW used in transport calculation is then $10 + 2.5 = 12.5$ km. The width of inner mooring is $2.5 + 5.1 = 7.6$ km, where 5.1 km is an effective width accounting for the shallowing bottom in the 10 km onshore of MN. The resulting cross-section area (600 m × 7.6 km) is equivalent to the area between 50 m and 650 m depth obtained by integrating the actual topography to 10 km onshore of the 650 m isobath.

**Table 2 of the original version listed the average and 1 standard deviation (we forgot to mention this in the caption, unfortunately). Now our error estimates are improved to also include standard error and all results are reported in the Table. Following up on reviewer 3's comment, we now also analyse summer 2016 and summer 2017 separately. New Table 2 is pasted below (see response to Reviewer 2 on details of the error calculations):**

**Table 2.** Volume transport calculations. Positive transport, $Q_p$ is directed northwest out of section, $Q_n$ is southeastward, and $Q$ is the total AW transport with $\Theta \geq 5$°C. $n$ is the degrees of freedom (daily data points divided by decorrelation time of 7 days). The values in square brackets are [$\pm\sigma$; $\pm$se], where $\sigma$ is the standard deviation, and se is the standard error (se $= \sigma/\sqrt{n}$). Additionally a total error estimate for $Q$ (see text) is given.

| Period | n | Transport (Sv) | | |
| --- | --- | --- | --- | --- |
| | | $Q_p$ | $Q_n$ | $Q$ |
| Annual | 66 | 2.1 [±1.3; ±0.2] | -0.1 [±0.1; ±0.0] | 2.0 ± 0.8 [±1.3; ±0.2] |
| Summer | 26 | 1.7 [±1.0; ±0.2] | -0.1 [±0.1; ±0.0] | 1.6 ± 0.7 [±0.9; ±0.2] |
| Summer-16 | 13 | 1.5 [±0.8; ±0.2] | -0.1 [±0.1; ±0.0] | 1.4 ± 0.6 [±0.7; ±0.2] |
| Summer-17 | 13 | 2.0 [±1.0; ±0.3] | -0.1 [±0.1; ±0.0] | 1.9 ± 0.7 [±0.9; ±0.3] |
| Winter | 13 | 3.0 [±1.9; ±0.5] | -0.0 [±0.1; ±0.0] | 2.9 ± 0.9 [±1.9; ±0.5] |

4) Transport of the top 50 m layer: The average volume transport cited in the abstract (2.8 Sv) seems not to include the top 50 m (line 213). I assume the reason to be that this layer is less saline than 35.17 on average (Figure 2) due to some admixture of water from the Norwegian Coastal Current, but isn't the fraction of Atlantic water in this layer still » 50% ? Using S=35.17 as a lower boundary for Atlantic water does not necessarily imply that you should use the same criterion for an upper boundary. If you want to retain this, it should in any case be better justified in the text.

**Re: This is a very good point. The reason we excluded the top layer was two fold: the low salinity layer as the reviewer noted but also, more importantly, the lack of measurements. We examined the upper layer from a freely accessible hydrological data set of the Nordic Seas (Bosse and Fer (2018) https://doi.org/10.21335/NMDC-1131411242). We showed details in our earlier response in discussion (figure R1). We infer that the fraction of AW exceeds 65% to 80% in the upper 50 m. While we do not include the top 50 m into the AW transport calculation, our sensitivity calculation (by extrapolating the uppermost measurement and assuming 100% AW fraction) is informative. In Section 6, we inserted:**

The crude estimate of the width of the slope current must be treated with caution. The sensitivity to the choice of mooring width is approximately linear. Reducing the total effective width by a factor of two, to 10 km, reduces the mean AW transport from 2.0 to 1.0 Sv. AW transports, on the other hand, are not sensitive to the definition of the AW temperature and vertical integration limits. Recalculating the transport using water with $\Theta \geq 3$ (instead of $\Theta \geq 5$) increases $Q$ by less than 0.1 Sv. While the upper layers are characterized by lower salinity water, the proportion of AW entrained into the upper 50 m should ideally be accounted for in the AW transport estimates. In the core of the slope current between MN and MW, salinity from the hydrographical atlas vertically averaged in the upper 50 m varies between 35.25 and 34.95 $\mathrm{g\,kg^{-1}}$ (not shown). Assuming shelf waters of salinity less than $34\,\mathrm{g\,kg^{-1}}$, the fraction of AW in the mixed water would exceed 65% to 80%. We limit our estimates at 50 m mainly because of lack of reliable current measurements. Including the upper 50 m by extending the uppermost available current measurement to the surface and assuming 100% AW fraction, increases the total mean transport by 0.3 Sv (from 2.0 Sv), well within the error estimates.

Details (points 5 to 15):

**Re: most of points 5 to 15 are minor points (typos etc.) and corrected as suggested (see our discussion response). In response to the relatively major comments (point 14) we improved the presentation of Fig A1, and (point 15) now incorporated the modelling part into the body of the manuscript.**

**Referee #2, Michael A. Spall**

This is a timely analysis of mooring data within the Norwegian Atlantic Slope Current on the eastern side of the Lofoten Basin. This region has been identified as a source of eddy kinetic energy and offshore eddy heat flux, which is important for the basinscale stratification and air-sea exchange. The analysis is fairly straightforward and I recommend that it be published subject to relatively minor revisions. There are a couple suggestions for additional analysis that, while not crucial, would provide more context for the results.

**Re. We thank the reviewer for the detailed reading and constructive comments. We addressed all comments as detailed below.**

Would it be possible to compare the transport in density and depth with that at the Svinoy section? If the transport there is barotropic, and here it is baroclinic, that implies that there has been some upwelling between these two stations, or a loss of transport in the deeper layers. If the transports are similar, can you tell if the isopycnals have risen or if there has been a water mass transformation between these two sections? I think a more complete comparison with that upstream section can reveal more about what has happened between these locations. Even if the years are different, maybe you can consider the seasonal cycle, which should be representative.

**Re. There is indeed substantial AW transformation between the Svinøy and Gimsøy sections. This was the subject of a previous paper (Bosse et al, JGR 2018). In response to this comment, we constructed annual mean T-S sections along the Svinøy and Gimsøy sections using the freely accessible dataset of the Nordic Seas (Bosse and Fer (2018) https://doi.org/10.21335/NMDC-1131411242), and inserted a new section (Section 7, Climatological structure and comparison with the Svinøy section) and a new figure (Fig 8). We inserted:**

**7 Climatological structure and comparison with the Svinøy section**

There is a substantial transformation of AW between the Svinøy (63°N) and Gimsøy (69°N) sections, discussed in detail by Bosse et al. (2018). Analyses on temperature/salinity space and in isopycnal layers showed that AW was progressively transformed to denser isopycnals. While the most important transformation occurred in the western part of the Lofoten Basin, lateral exchanges generated by instabilities of the slope current substantially modified the characteristics of the AW transported from the Svinøy to Gimsøy section. A climatological view of the hydrography in the Svinøy and Gimsøy sections shows the important cooling and freshening of AW (Fig. 8). As the AW is modified, isopycnals with potential density anomaly $\sigma_0$ less than 27.7 kg m$^{-3}$ rise. At the core of the slope current, the displacement of the 27.5 isopycnal reaches 150 m, switching from being located below the AW core to above. This is also where the largest spiciness injection - an indicator of water mass transformation by diapycnal mixing - by vertical mixing was reported (Bosse et al., 2018). Deeper isopycnals sink from Svinøy to Gimsøy, which could be related to the intermediate waters subducted along the Mohn Ridge front and AW transformations in the Lofoten Basin, decreasing the stratification in the AW pycnocline. As a result of winter mixing driven by intense air-sea fluxes, the AW pycnocline in the Lofoten Basin is more diffuse and deeper at around 800 m (vs. 500 m farther south). The cross-slope temperature and salinity gradients across the slope current also exhibit a different structure, suggesting different contributions to geostrophic currents (via thermal wind balance) and a change with latitude in baroclinicity of the slope current (Fig. 9).

[Figure]

**Figure 8.** Mean (a-c) $\Theta$ and (b-d) $S_A$ distribution along the Gimsøy and Svinøy sections obtained from the Nordic Seas data set (Bosse and Fer, 2018). Contour interval is 1°C for $\Theta$ and 0.1 g kg$^{-1}$ for $S_A$. Salinity is saturated at 34.4 g kg$^{-1}$, but minimum values are 33.0 at Svinøy and 33.9 at Gimsøy. Isopycnals (potential density anomaly referenced to surface pressure, $\sigma_0$, black) are drawn at 0.2 kg m$^{-3}$) interval to 27 kg m$^{-3}$, followed by 26.5 and 26 kg m$^{-3}$ for shelf waters. An inset map for each section shows the profiles used, located within 25 km distance from the sections. Distance is referenced to the 500 m isobath.

Can the authors provide error bars for the velocity and transport estimates?

**Re. We now provide error bars for the velocity and transport estimates: for winter profiles in Fig 4 (for both the temperature and velocity profiles), for monthly averaged AW transport in Fig 6b, for the annual average in T-binned histogram of Fig. 7, and in Table 2 where we list the average transports. In the figures, the error bars are based on the standard error. In addition to the standard error, we estimate a representative error for the transport (including error estimate from the width and depth-averaged current). Table 2 is updated with errors calculated for each analysis period (annual, summer and winter; and summer 2016 and summer 2017, separately; revised Table 2 is pasted in response to point#3 of reviewer#1).**

**In Section 6, we inserted:**

Together with the temporal averages and one standard deviation, $\sigma$, we also report the standard error of the mean and a representative total transport error estimate. The standard error is calculated as $se = \sigma/\sqrt{n}$, using a degrees of freedom ($n$) taking into account the decorrelation time scale of 7 days (Sect. 5). We calculate a representative transport error estimate, for winter, summer and annual data points separately, accounting for the time variability in statistics. At each mooring, we assume root-mean-squared errors of about 20% (4 km) in the effective width and $0.05\,\mathrm{m\,s^{-1}}$ in depth averaged current (corresponding to $30\,\mathrm{m^2\,s^{-1}}$ of transport density). A simple calculation using these figures, ignoring the statistics, would lead to an error of 0.12 Sv. Using the mean and $\sigma$ of observed transport density (for winter, summer and all data separately), we generate 100 random data points from a normal distribution and calculate the transport (without imposed error) using 20.1 km width. The distribution of transport is approximately normal in each season, and this assumed distribution for error analysis is justified. We then generate 100 values for transport density and width from a random distribution with imposed errors, and calculate the total transport (with error). The root-mean-squared value of the difference between transport values with and without error from this 100-point realization gives one error estimate. We draw 1000 bootstrap error estimates and average them to obtain the reported error. The transport error is 0.8 Sv for the annual average, 0.7 Sv for summer and 0.9 Sv for winter averages. This is typically less than the standard deviation and 3-4 times the standard error (Table 2).

Introduction:

You might also reference Clark and Straneo (Observations of Water, Mass Transformation and Eddies in the Lofoten Basin of the Nordic Seas, JPO, 2015).

**Re: Done.**

I had a 2010 paper in Ocean Modeling that would be more appropriate to reference than the 2010 JPO paper as it addresses the lateral eddy heat flux in (an idealized) Lofoten Basin (Spall,Non-local topographic influences on deep convection: An idealized model for the Nordic Seas, Ocean Modeling, 32, 72-85).

**Re. Thank you for point this reference out. Changed as suggested.**

lines 120-124: It should be possible to quantify the source of the increased vertical shear, or at least break it down into temperature and haline contributions via thermal wind.

**Re. In the new Section 7, we now present the vertical shear of geostrophic current perpendicular to Gimsøy and Svinøy sections derived from the annual mean hydrography described above. We also break it down to into temperature and haline contributions (new Fig 9). Text from Section 7 is pasted below together with Fig 9.**

The cross-slope gradients are relatively weaker at Gimsøy compared to Svinøy section, and so are the temperature contribution (positive at the slope, negative on the shelf) and haline contribution (negative at the slope, positive on the shelf) to the geostrophic shear (Fig. 9). Furthermore, the coastal current core - identified by the positive shear driven by salinity on the shelf - interacts more strongly with the slope current at Gimsøy. This can be explained by the steeper slope of the Lofoten Escarpment, which has a stronger control on the mean position of the slope current. Note that the broader region of isopycnal gradients at Svinøy does not necessarily imply a broader current, but could result from a more variable position linked to a weaker topographical control by the steepness of the slope.

To further compare the baroclinicity of the slope current at these two locations, we vertically integrated the different contributions to the geostrophic shear with a level of no motion at the bottom (geostrophic velocity contours in Fig. 9). The baroclinicity of the slope current indeed increases with latitude: poleward geostrophic currents exceed $0.6 \, \text{m s}^{-1}$ at Gimsøy compared to about $0.4 \, \text{m s}^{-1}$ at Svinøy, despite the stronger contribution from vertically-integrated shear due to temperature ($0.75 \, \text{m s}^{-1}$ at Svinøy vs. $0.56 \, \text{m s}^{-1}$ at Gimsøy). A strong negative shear due to salinity counter-balances the thermally-driven geostrophic shear of the current at Svinøy (reaching $-0.31 \, \text{m s}^{-1}$ integrated from bottom to 150 m, $-0.25 \, \text{m s}^{-1}$ to the surface). At Gimsøy, this value reaches only $-0.12 \, \text{m s}^{-1}$ from the bottom to 250 m, and becomes insignificant when integrated to the surface. This suggests that the cross-slope salinity gradient is important for the baroclinicity of the slope current, even in a region where temperature accounts for most of the density variations. Changes in the baroclinicity of the slope current can thus be expected following the recent AW freshening observed in the Nordic Seas (Mork et al., 2019).

[Figure]

**Figure 9.** Vertical shear from thermal wind balance for (upper row) Gimsøy and (lower row) Svinøy sections using the annual-mean hydrography shown in Fig. 8. Panels (a) and (d) are the total geostrophic shear, (b) and (e) are the thermal contribution, and (c) and (f) are the haline contribution to shear. Vertically-integrated shear is also contoured (blue: negative; red: positive values). Distance is referenced to the 500 m isobath. Isolines are drawn at $0.1 \, \text{kg m}^{-3}$ for $\sigma_0$ (down to $27 \, \text{kg m}^{-3}$, and with additional 26.5 and $26 \, \text{kg m}^{-3}$ contours), 1°C for $\Theta$, 0.2 $\text{g kg}^{-1}$ for $S_A$, and $0.1 \, \text{m s}^{-1}$ for vertically-integrated shear.

line 157: It might be useful to provide a scaling for the expected response to changes in the wind stress. One could calculate the onshore Ekman transport, downward deflection of the isopycnals, and the geostrophic response. The paper by Choboter et al. (2011, Exact Solutions of Wind-Driven Coastal Upwelling and Downwelling over Sloping Topography, JPO, 41, 1277-1295) provides analytic solutions but you might be able to do something useful just with simple scaling.

**Re. Thank you for pointing out this reference. We could not easily find a way to estimate the vertical excursion using the analytical solutions or the scaling in this paper. A typical along wind speed event of 10 cm/s (Fig 10a) gives stress of 0.14 Pa and Ekman transport per unit along shelf distance of 1 m$^2$/s. If one assumes an Ekman layer of 50 m, and processes occurring at shelf of 250 m (i.e. H = 200 m), the scales following Choboter et al. are on the order of 1 m/s for the along slope velocity and less than 1 cm/s for the cross-slope velocity scale (in the interior below the Ekman layer). No action taken.**

line 191: It seems likely that the transport variability is due to the current meandering outside the moorings (rather than a change in the along-slope transport), but this isn't explicitly mentioned..

**Re: Agreed. We now mention this point.**

Figure 7: I found this to be the most surprising part of the paper. Any ideas why there is more warm water in winter than in summer? When/where was this water last exposed to the atmosphere? Was this subducted in the previous summer? If you see the same phase at Svinoy, which is O(1000 km) upstream, that would argue against it simply being advected along the slope. I think some more discussion around this finding would be helpful. The penetration of AW down to 650 m depth is likely related to that being the sill depth upstream.

**Re. In our response during discussion we presented the time series of depth-averaged along-isobath current and temperature at Svinøy and the Lofoten moorings, MN and MW (Fig R3, not repeated here). A seasonal signal in temperature is clearly observed at Svinøy. The pattern is similar at Gimsøy, and the winter temperature anomaly is larger. While there may be issues from the lack of temperature data in the upper layers in our moorings, overall, we interpret the largest warm water transport in winter as a consequence of the annual cycle of depth-averaged temperature coinciding with the time of strongest currents (winter).**

line 232: BC and BT were also calculated from a high resolution mooring array in Spall et al. (2008).

**Re: We now include this in our list of examples.**

line 238-239: CHECK!NOT 1 MONTH?

**Re: corrected.**

lines 242-245: This justification is not very convincing, I suggest deleting it.

**Re: Deleted as suggested.**

line 340: Magenta does not stand out compared to the colorbar, I suggest using a different color to mark the line.

**Re: We improved this figure.**

**Referee #3, Anonymous**

Overview and general recommendation:

The manuscript describes the outcome of a mooring effort, carried out between Jun 2016 and August 2017 across the Norwegian Atlantic Slope Current off the Lofoten Islands at the so-called Lofoten Escarpment. The authors exploit the data from a mooring array that consisted of three deep sea moorings. Two of them, moorings MN and MW, were located about 6 km apart from each other across the slope current. A third mooring, MS was located almost 30 km further upstream close to the Gimsøy hydrographic repeat section. A fourth mooring, MB, was located in the interior of the Lofoten Basin and was not part of the analysis. The authors use the mooring records, mainly velocity data obtained from Longranger ADCPs as well as T/S information from MicroCATs or temperature loggers, to address the Atlantic Water (AW) layer within the Norwegian Atlantic Slope Current that is captured by the moorings. While there are already descriptions of the Slope Current from the sections located upstream, the authors state that they provide the first mooring based description for the Gimsøy region off the Lofoten Islands. The authors describe the general nature of the velocity structure in the upper water column and find the strongest velocities in the winter period. This timing coincides with the time of the warmest temperatures observed in the AW layer. The authors furthermore infer transport time series for the two moorings MN and MW, explain their choice of a respective area over which the transport is calculated and finally quantify the volume transport for the AW layer. The authors furthermore address the forcing and find a correspondence between the along-stream wind forcing and the along-stream current component. Finally, the authors infer energy conversion rates from the mooring records, in particular baroclinic and barotropic conversion rates that describe the transfer of mean potential energy into eddy kinetic energy and the transfer of mean kinetic to eddy kinetic energy. The baroclinic conversion rate can only be estimated for the first three months of the deployment period due to otherwise missing data. The authors find conversion rates with magnitudes similar to estimates inferred for the East Greenland Current and the West Spitsbergen Current. Due to limitations in the mooring data set the authors have considered output from a high-resolution ROMS model. The respective analysis is part of an appendix to the paper. Therein, the authors aim at verifying how representative the mooring-derived energy conversion rates actually are. They conclude from the model analysis that the baroclinic energy conversion dominates over the barotropic energy conversion.

**Re: We thank the reviewer for the detailed reading and constructive comments. We addressed all comments as detailed below.**

In general, the paper is written well enough. But I personally found it sometimes a bit tiring to read all the abbreviations. This is probably a matter of personal taste. I did wonder, however, why the model analysis was somewhat "hidden" in the appendix. The authors draw important conclusions from this model analysis. Any reader might easily miss the respective discussion by simply ignoring to read the appendix. Therefore, I think, this analysis deserves to be built into the main text.

**Re. Thank you for this suggestion. We agree. We integrated the Appendix to the main body of the revised manuscript. We also removed some abbreviations.**

The study of the authors contributes to improving the knowledge of one of the major currents transferring the warm and saline Atlantic Water towards the Arctic. I find that the manuscript addresses interesting scientific outcome on the nature of this current off the Lofoten Islands that is of interest to the readers of OS. The figures are generally of high quality. However, I partly missed information regarding the methods applied to the mooring time series. For example, it was several times mentioned that the mooring succumbed to "knock-down" events. But how these events were eliminated from the

data remained unclear. There are other minor requests for clarification that I think will help to improve the manuscript further. Therefore, I recommend a minor revision of the manuscript.

**Re. Thank you for your assessment of our manuscript. We improved the description of methods and applied various clarifications as detailed below.**

My detailed comments are given below:

Page 1, line 25: the statement that "the front current is relatively poorly known" somehow contradicts the statements that follow in the next sentences. Therein, the authors quote several studies that provide transport estimates for the front current for various location. It might help to clarify what exactly is "relatively poorly known".

**Re. We replaced this opening sentence with "The front current, which is not addressed in this study, has not been measured in detail using current meter arrays, but geostrophic transport estimates are available from hydrography."**

Page 3, line 36: please highlight the location of the Lofoten Escarpment in Figure 1 by adding a respective label. Same sentence starting with "there might. . .": there is a word missing, "be"?

**Re. Done**

Page 3, line 59: please add something like "based on the mooring records" at the end of the sentences

**Re. Done**

Page 4, Table 1: as there are different styles/cultures to write down dates, I suggest to write months using letters like May, Jun, Sep. This avoids that people mix up days and months.

**Re. Done**

Page 4, line 62: it remains unclear what kind of manuscript "Fer (2020)" actually is or where it can be assessed. The respective reference does not provide any relevant information. So, at present, any reader is not able to locate information on the data set other than the one mentioned here. Same holds for page 5, lines 84/85, where the same reference is mentioned.

**Re. Thanks for noticing this. Unfortunately, Fer (2020) was not formatted properly. It is the mooring data set together with a detailed data report, openly accessible with CC BY 4.0 license from the Norwegian Marine Data Centre. Neither the URL, DOI or the fact that this is a data set, was apparent in the citation (because of the formatting). We did not notice this. We now corrected it and also updated the data availability statement with the link.**

*Data availability.* Mooring data used in this analysis are available from Fer (2020), from https://doi.org/10.21335/NMDC-1664980441 through the Norwegian Marine Data Centre with Creative Commons Attribution 4.0 International License. The data set of the Nordic Seas (Bosse and Fer, 2018) is available from https://doi.org/10.21335/NMDC-1131411242, through the Norwegian Marine Data Centre with Creative Commons Attribution 4.0 International License. Other environmental data are obtained using Copernicus Climate Change Service (C3S) (2017), and European Centre for Medium-range Weather Forecast (ECMWF) (2011). Sea level anomaly data are obtained from the E.U. Copernicus Marine Service Information, product SEALEVEL_GLO_PHY_L4_REP_OBSERVATIONS_008_047.

Page 4, line 80: previously, it was said that the used ADCPs were of type RDI 75 kHz Longranger. Now, they are addressed as RDI 75 kHz Sentinel Workhorse. To my knowledge, such a device operating at 75 kHz does not exist. According to the Teledyne-RDI web page, ADCPs of type Sentinel operate at frequencies >= 300 kHz and thus have a much shorter range than the Longranger ADCPs. Please, clarify.

**Re. The reviewer is correct. "75 kHz Sentinel" is a mistake and now corrected. In MN, MS and MW we used only Longranger ADCPs. (Additional 300 kHz Sentinels were also deployed in MB, but not reported in this paper.)**

Page 5, line 88: please, provide more information on the observed knock-down events, e.g. how often did they occur, how deep did the moorings descend, and how was this effect eliminated from the considered data ? Very much later (page 15), it is mentioned in the text that the data set was actually interpolated and gridded. This information and related specifics are missing here.

**Re. We inserted the following description in the Data (2.1 Moorings) section. More details can be found in the data report following the openly accessible data set:**

Substantial vertical displacements ("knock-down") of the mooring line occurred in response to strong current events at MW and MB (not reported here). At MW, the vertical displacements recorded by the uppermost pressure sensor at 75 m target depth was 7 m (50 percentile), 15 m (80 percentile, corresponding to a total duration of about 3 months) and 68 m (97 percentile, corresponding to events with a total duration of 2 weeks). The vertical displacements were reduced by approximately a factor of two at the level of the ADCP flotation at 740 m depth. The velocity measurements from the ADCPs installed in the bottom units at MN and MS were relatively unaffected by the mooring motion (typical vertical displacements associated with knock-down were less than 1 m with a 97 percentile value of 2 m). Overall, the moorings were equipped with several pressure sensors which we used to approximate the depth of temperature, salinity and current measurements in the water column.

A data set was prepared after correcting for mooring knock-downs. Data from all instruments were first averaged into one hour intervals (if the sampling rate was faster) and then interpolated to a common 1-hour time array. Time series of instrument depth were constructed at each time and for each instrument using vertical interpolation of the known target depth (of instruments with pressure sensor) and the measured pressure to the target depths of all instruments. Hourly profiles of temperature, salinity and horizontal current were then vertically interpolated to a uniform 10-m vertical resolution. Data gaps at a given vertical level were typically caused by mooring knock-down or lack of acoustic scatterers for Doppler velocity measurements. At MW, velocity measurements were relatively limited in the vertical. The data gap in the time series was 18% at 250 m depth, reaching 60% at 200 m. The vertical extent of temperature measurements at MW was better: temporal gap at 90 m was only 20%, increasing to 70% at 80 m. The missing velocity data at MN was 35% at 150 m, increasing to 50% at 80 m. A depth level with a data coverage less than 30% of the total measurement duration was excluded from the data set.

Page 5, line 100: please, refer here to the Copernicus Marine Environmental Monitoring Service (CMEMS) as the data provider, since there are still a number of papers out that still claim AVISO to be the data provider. Use of CMEMS data furthermore expects if not requires a proper credit of their data use, which is missing in this manuscript. My guess is that ECMWF expects something similar. Finally, as EKE is not a property provided as part of the used data product, how is EKE defined here ? Did you just consider the provided geostrophic anomalies ? The data set provides both, anomalies and absolute velocities. The present text is not clear enough on what kind of velocity fluctuations actually been used.

**Re. We now refer to E.U. Copernicus Climate Change Service and E.U. Copernicus Marine Service Information as the data provider, and properly cite ERA-5 and ERA-Interim. We calculated the EKE using the geostrophic current anomalies obtained from the sea-level anomaly. This is now clarified in the revised version. In new Section 2.2 we inserted the following (also updated Data Availablity statement (pasted above) and the reference list):**

**2.2 Other data**

Atmospheric forcing was obtained from European Centre for Medium-range Weather Forecast (ECMWF) (2011) ERA-Interim (Dee et al., 2011) reanalysis over the historical time period from 1979 to 2018, and from higher resolution ERA-5 reanalysis (Copernicus Climate Change Service (C3S), 2017) over the mooring observation period. Surface net fluxes $Q_{net}$ (downward positive) were computed as the sum of net shortwave and longwave contributions and latent and sensible heat fluxes. Time series of fluxes were extracted at the nearest grid point from mooring sites. We calculated the geostrophic EKE ($EKE_g$) using the surface geostrophic velocity anomalies obtained from sea-level anomaly from the multimission altimeter satellite gridded sea surface height observations distributed by E.U. Copernicus Marine Service Information.

Page 5, line 116: please, add "2017" after "March".

**Re. Done**

page 7, line 135: it looks like the cross-component was also quite high in spring. In the upper part of the water column (Figure 4b) other seasons seem to be higher than winter or are of comparable magnitude. Could you comment on that?

**Re. We revised this part as:**

properties flowing into the Nordic Seas. Cross-slope component was weak (typically $\pm 0.02$ m s$^{-1}$) and increased in spring and winter, with largest 200-600 m depth-averaged values in winter (0.05 m s$^{-1}$ at MW) and an increased variability with depth (Fig. 4b, d). In the upper part of the water column at MW, averaged cross-slope velocities in fall exceeded the winter values. This is consistent with the increased $EKE_g$ at MW location, calculated from satellite measurements in November 2016 (Fig. 3c). In deep layers (>900 m) at MW, barotropic currents were between 0.05 and 0.10 m s$^{-1}$.

Page 10, line 171: you could either repeat the separation distance of 6 km here, or otherwise provide readers with the size of the Rossby deformation radius at this location

**Re. We repeat the separation distance (note that this is now reduced to 5 km; see response to reviewer 1).**

page 10, lines 176/177: from figures 4 and 5 it is obvious that there isn't any velocity data at depth < 200m at mooring MW. At times, there are data missing as deep as 300 m. Also MN does show data gaps for z < 200m. So, please, clarify how it is possible to infer the transport for the 50-650m range.

**Re. The shallowest available measurement is extended upward to 50 m. This is now clarified.**

**At MW, velocity measurements are limited in the vertical. The data gap is 18% at 250 m depth and rapidly increases to 60% and 85% at 200 and 190 m. Temperature at MW is better covered with 20% gap at 90 m, increasing to 70% at 80 m. Velocity measurements at MN have 35% gap at 150 m, increasing to 50% at 80 m. This information is now provided in Section 2.1 (pasted above in response to comment "Page 5, line 88").**

Page 10, line 179: how did you treat the temperature information outside the mooring array or during those times, when there wasn't any temperature information for mooring MW? Did you consider the depth of the 5C isotherm to be constant across the entire width of the area used for calculating transports? Please, clarify as well.

**Re. For full-duration transport calculations, we used the temperature record from MW, which is available for the entire deployment and is well-resolved in the vertical. We extended the uppermost**

**temperature measurement to 50 m when there were gaps in the data (typically above 90 m depth, see above). These points are now clarified.**

page 10, line 190: please, clarify; relative to what reference level? Same line: "currents peak" or "the current peaks"?

**Re. clarified and corrected: relative to surface pressure; current peaks**

Page 11, line 200: as the summer season is covered twice, is "summer" meant here as the average of both summer seasons? Was there any difference between the two summers? One might guess so by looking at Figure 6.

**Re. Yes, we averaged both summers. We now also report and discuss the differences between summer 2016 and 2017. We do not show the mean profiles separately because Fig 4 is too crowded. We inserted:**

The summer profiles in Fig. 4 are averages over summers of 2016 and 2017. When averaged separately (not shown), temperature profiles at MW are very similar, equal to within 0.5°C in the upper 600 m and identical in deeper layers. At MW, $u$ was 0.01 m s$^{-1}$ larger below 300 m (a small barotropic increase) in summer 2017, and shear was stronger in the upper 300 m, increasing by 0.06 m s$^{-1}$ to 200 m depth. At MN, summer-average profiles of $u$ in the bottom 250 m were identical in 2016 and 2017, but shear was stronger higher in the water column (above 400 m) in summer 2017, with $u$ increasing by an additional 0.10 m s$^{-1}$ to 200 m depth. This implies a substantial inter-annual variability in the upper 300 – 400 m which cannot be resolved by our limited times series.

**Transport is stronger in summer 2017 relative to 2016. Average values, standard deviation and standard errors are now listed separately for each summer in Table 2 (pasted below)**

page 12, Table 2: Please write Q_N and Q_S in the same way as it is used in the text and in the table, i.e. with small letters for "N" and "S". As will also be my question regarding Figure 7: as "annual" refers to the entire time series, and as this comprises two summer seasons, does this enter the uncertainties? Or asked differently, what is included in the uncertainties mentioned here?

**Re. Corrected (we are now using p(ositive) and n(egative)). The averaging durations, standard deviations, degrees of freedom and standard errors are now calculated and clarified in the table and in the text. New Table 2 is pasted below.**

**Table 2.** Volume transport calculations. Positive transport, $Q_p$ is directed northwest out of section, $Q_n$ is southeastward, and $Q$ is the total AW transport with $\Theta \geq 5°C$. $n$ is the degrees of freedom (daily data points divided by decorrelation time of 7 days). The values in square brackets are $[\pm\sigma; \pm se]$, where $\sigma$ is the standard deviation, and se is the standard error (se $= \sigma/\sqrt{n}$). Additionally a total error estimate for $Q$ (see text) is given.

| Period | n | Transport (Sv) | | |
|---|---|---|---|---|
| | | $Q_p$ | $Q_n$ | $Q$ |
| Annual | 66 | 2.1 [$\pm$1.3; $\pm$0.2] | -0.1 [$\pm$0.1; $\pm$0.0] | 2.0 $\pm$ 0.8 [$\pm$1.3; $\pm$0.2] |
| Summer | 26 | 1.7 [$\pm$1.0; $\pm$0.2] | -0.1 [$\pm$0.1; $\pm$0.0] | 1.6 $\pm$ 0.7 [$\pm$0.9; $\pm$0.2] |
| Summer-16 | 13 | 1.5 [$\pm$0.8; $\pm$0.2] | -0.1 [$\pm$0.1; $\pm$0.0] | 1.4 $\pm$ 0.6 [$\pm$0.7; $\pm$0.2] |
| Summer-17 | 13 | 2.0 [$\pm$1.0; $\pm$0.3] | -0.1 [$\pm$0.1; $\pm$0.0] | 1.9 $\pm$ 0.7 [$\pm$0.9; $\pm$0.3] |
| Winter | 13 | 3.0 [$\pm$1.9; $\pm$0.5] | -0.0 [$\pm$0.1; $\pm$0.0] | 2.9 $\pm$ 0.9 [$\pm$1.9; $\pm$0.5] |

Page 13, equation 1: if EKE is inferred from along-stream and across-stream velocities, it makes sense to keep the previously inferred terms u_a and u_x. Here and later in the text, the authors switch to u and v.

**Re. We now use u and v (along and across isobath components) throughout.**

Page 13, line 241: please, provide a reference.

**Re. Inserted Spall et al. (2008).**

Page 14, lines 243/244: these lines need fixing. Furthermore, equation (2) does not contain a rho_0, which is part of equation (3), but a rho', which is not introduced. What reference density was used ? Shouldn't the right term of equation (2) be negative?

**Re. Corrected, and all these points now clarified. We used $\rho_0$ = 1027 kg m$^{-3}$. The sign of Eq(2) is correct in the convention we described (it can also be obtained by simplifying Eq. A1; note the sign of the last term.).**

Page 15, lines 256: four times use of the word 'obtain'

**Re. Improved the text.**

page 15, lines 260, sentence starting with "The conversion rates calculated from. . .": please remove this entire sentence and the following as the information is identical to the one given in lines 279ff. There, it fits much better.

**Re. Done.**

Page 15, line 269: the statement that the estimates from the Fram Strait are comparable to the Lofoten Escarpment is a bit tricky. The former values are O(100) m^2/s^2, the latter values are O(10^-4) W/m^3. So, please, make the comparability more obvious to the reader.

**Re. We revised and clarified this part as follows:**

Average barotropic conversion rate (averaged over both moorings, over multiple levels and over 14 months) was $(0.3\pm0.2)\times 10^{-4}$ W m$^{-3}$. Maximum value reached $1.2\times 10^{-4}$ W m$^{-3}$. The baroclinic conversion rate (only available in the summer for the first three months of the mooring period) was comparable, $(0.4\pm0.6)\times 10^{-4}$ with a maximum of $1.7\times 10^{-4}$ W m$^{-3}$. For reference, a conversion rate of $10^{-4}$ W m$^{-3}$ for 1 day accounts for $\rho_0$ EKE of $O(10)$ J m$^{-3}$, or EKE of $O(100)\times 10^{-4}$ m$^2$ s$^{-2}$.

Observed EKE and the conversion rates at the Lofoten Escarpment can be compared to other relevant observations. In Fram Strait von Appen et al. (2016) analyzed 12 year long time series from moorings with focus on the West Spitsbergen Current. EKE at 75 m depth was $50\times10^{-4}$ m$^2$ s$^{-2}$ in summer, and increased to $200\times10^{-4}$ m$^2$ s$^{-2}$ in winter. At 250 m depth the magnitude was approximately reduced to half. These values, overall, are similar to the EKE at the Lofoten slope. In terms of baroclinic and barotropic conversion rates, the two sites are also comparable: BT was on the order $0.1\times 10^{-4}$ W m$^{-3}$, and BC at 75 m was $0.5\times 10^{-4}$ W m$^{-3}$ in summer, increasing to $1.5\times 10^{-4}$ W m$^{-3}$ in winter. Summer mean and maximum values are identical (within measurement uncertainties) to the corresponding values from our observations at 400 m in summer.

Page 15, line 270: please introduce WSC

**Re. We decided to remove this abbreviation because it is used only a couple of times.**

page 16, line 298: there is a word missing at the end of the line.

**Re. Corrected.**

Figures:

Figure 1. Labels like "Norway" in Figure 1a and "NO" in Figure 1b are really hard to see. Think about adding a text label highlighting the location of the Lofoten Escarpment. The unit in the EKE colorbar should read 10ˆ-4 mˆ2sˆ-2, not 10ˆ4 mˆ2sˆ-2.

**Re. Improved and corrected as suggested.**

Figure A1. Both subplots lack a frame, at least in my printed version. Also the grid is almost invisible in the printed version. Maybe the authors can improve that. To the southwest of the red box, there is something like an arc-like pattern of very small-scale features in Fig A1a that look totally different from the remaining parts of the plot. What causes this?

**Re. The new Fig (11) is much improved as shown below:**

[revised manuscript text omitted]